# Unconventional Hall response in the quantum limit of HfTe$_5$

S. Galeski [1✉], X. Zhao[2,3,4], R. Wawrzyńczak[1], T. Meng [5], T. Förster[6], P. M. Lozano[7,8], S. Honnali[1], N. Lamba[1], T. Ehmcke [5], A. Markou [1], Q. Li. [7,8], G. Gu[8], W. Zhu[2,4], J. Wosnitza[6,9], C. Felser[1], G. F. Chen [2,3,4] & J. Gooth [1,9✉]

Interacting electrons confined to their lowest Landau level in a high magnetic field can form a variety of correlated states, some of which manifest themselves in a Hall effect. Although such states have been predicted to occur in three-dimensional semimetals, a corresponding Hall response has not yet been experimentally observed. Here, we report the observation of an unconventional Hall response in the quantum limit of the bulk semimetal HfTe$_5$, adjacent to the three-dimensional quantum Hall effect of a single electron band at low magnetic fields. The additional plateau-like feature in the Hall conductivity of the lowest Landau level is accompanied by a Shubnikov-de Haas minimum in the longitudinal electrical resistivity and its magnitude relates as 3/5 to the height of the last plateau of the three-dimensional quantum Hall effect. Our findings are consistent with strong electron-electron interactions, stabilizing an unconventional variant of the Hall effect in a three-dimensional material in the quantum limit.

[1] Max Planck Institute for Chemical Physics of Solids, Nöthnitzer Straße 40, 01187 Dresden, Germany. [2] Institute of Physics and Beijing National Laboratory for Condensed Matter Physics, Chinese Academy of Sciences, 100190 Beijing, China. [3] Songshan Lake Materials Laboratory, 523808 Dongguan, Guangdong, China. [4] School of Physics Science, University of Chinese Academy of Sciences, 100049 Beijing, China. [5] Institute of Theoretical Physics and Würzburg-Dresden Cluster of Excellence ct.qmat, Technische Universität Dresden, 01062 Dresden, Germany. [6] Hochfeld-Magnetlabor Dresden (HLD-EMFL) and Würzburg-Dresden Cluster of Excellence ct.qmat, Helmholtz-Zentrum Dresden-Rossendorf, 01328 Dresden, Germany. [7] Department of Physics and Astronomy, Stony Brook University, Stony Brook, NY 11794-3800, USA. [8] Condensed Matter Physics and Materials Science Department, Brookhaven National Laboratory, Upton, NY, USA. [9] Institut für Festkörper- und Materialphysik, Technische Universität Dresden, 01062 Dresden, Germany. ✉email: stanislaw.galeski@cpfs.mpg.de; johannes.gooth@cpfs.mpg.de

Applying a strong magnetic field to an electron gas confines the electrons motion in cyclotron orbits with a set of discrete eigenenergies—the Landau levels. In two-dimensional (2D) systems, this quantization leads to a fully gapped energy spectrum and to the emergence of the quantum Hall effect (QHE)[1]. In the limit where only the lowest Landau level (LLL) is occupied (the so-called quantum limit), electron–electron interactions can play a significant role, leading to the appearance of correlated states, such as the fractional quantum Hall effect[2]. In contrast, the Landau level spectrum of a three-dimensional (3D) electron gas is not fully gapped and becomes like that of a one-dimensional system. As a consequence, the electrons can still move along the field direction, which in turn destroys the quantization of the Hall effect[3–5]. Nevertheless, it has been predicted that a generalized version of the QHE can emerge in 3D electron systems that exhibit a periodically modulated superstructure[6–8]. Analogous to as in two dimensions, in the vicinity of the quantum limit, 3D electron systems are also prone to form a variety of correlated electron states, including Luttinger liquids; charge, spin and valley density waves; excitonic insulators; Hall and Wigner crystals; or staging transitions in the case of highly anisotropic layered systems[3,4,6,9–13]. It has been theoretically pointed out that some of these states are related to quantum Hall physics in three dimensions and likewise could manifest themselves in a Hall response that should be observable in the quantum limit of 3D semimetals[10,13,14].

Inspired by these ideas, the possibility of finding a 3D QHE has been explored in several material systems. For example, signatures of the integer quantum Hall effect (IQHE) have been found in quasi-2D semiconducting multilayer lattices[15], Bechgaards salts[16,17], $\eta$-$Mo_4O_{11}$[18], $n$-doped $Bi_2Te_3$[19], and $EuMnBi_2$[20], in which the layered crystal structure itself supplies the stack of 2D systems. Very recently, the QHE has also been observed in 3D graphite films[5], bulk $ZrTe_5$[21], and $HfTe_5$ samples[22]. In graphite, the imposed periodic superstructure has been attributed to the formation of standing electron waves. In $ZrTe_5$ and $HfTe_5$, the IQHE was originally believed to arise from a charge density wave (CDW), due to the scaling of plateau height with the Fermi wavevector. This scenario is, however, in contrast with thermodynamic and thermoelectric measurements on $ZrTe_5$ that did not reveal any signatures of a field-induced CDW transition. Instead, it was proposed that $ZrTe_5$ should be considered a stack of weakly interacting Dirac 2DEGs with the plateau height scaling originating from the interplay of small carrier density and the peculiarities of Landau quantization of the Dirac dispersion[23]. In parallel to the search for the 3D QHE, there has been a long-standing experimental effort to observe field-induced correlated states in the quantum limit of three-dimensional materials.

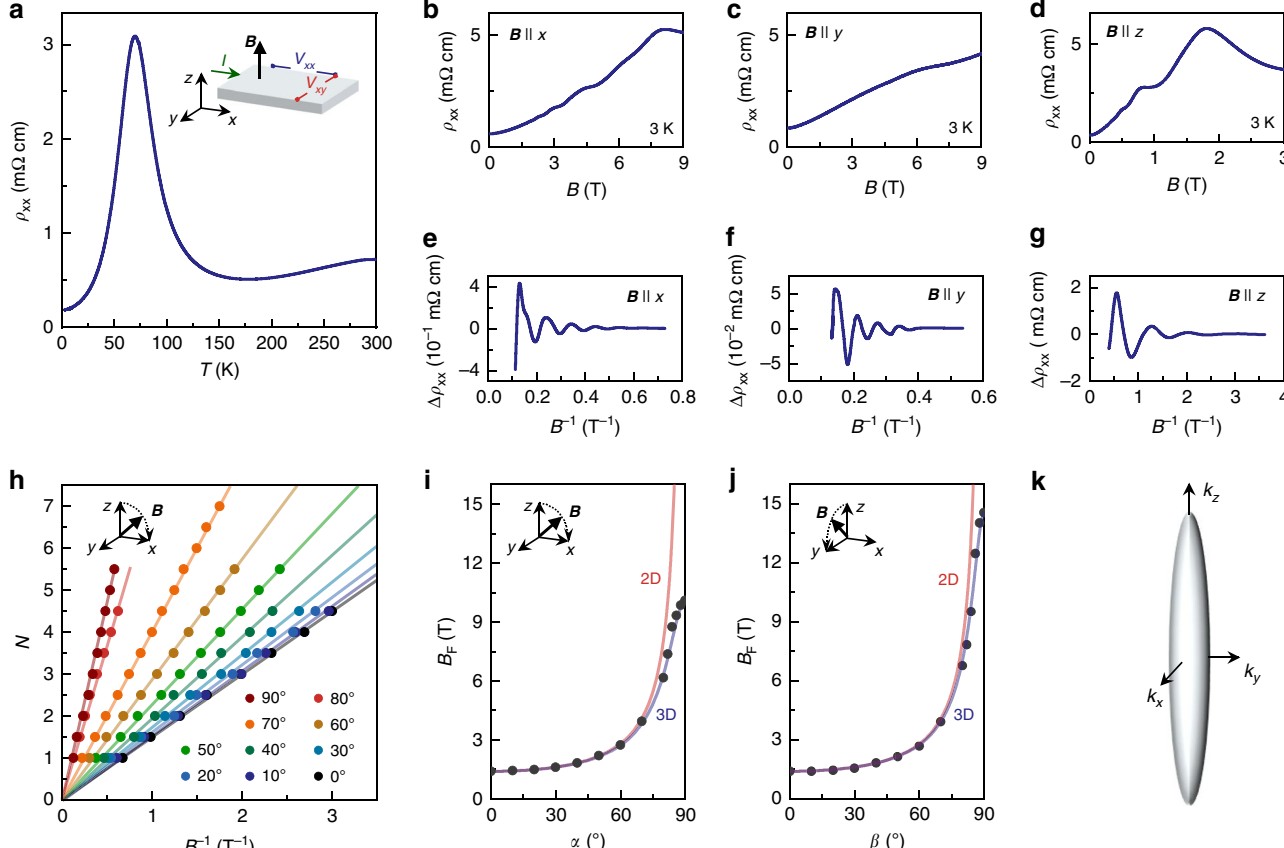

**Fig. 1 Three-dimensional morphology of the Fermi surface in HfTe₅. a** Longitudinal electrical resistivity $\rho_{xx}$ as a function of temperature $T$ at zero magnetic field. Inset: Sketch of the measurement configuration in the three spatial directions $x$, $y$, and $z$. The bias current $I$ is applied along $x$ and the magnetic field **B** along $y$. The corresponding voltage responses are measured in $x$- ($V_{xx}$) and in $y$ direction ($V_{xy}$). **b** $\rho_{xx}$ as a function of **B** at 3 K with **B** applied along $x$, **c** along $y$ and **d** along the $z$ direction. **e** Variation of the longitudinal electrical resistivity $\Delta\rho_{xx}$ as a function of **B** at 3 K with **B** applied along $x$, **f**, along $y$, and **g** along the $z$ direction. **h** Landau-index fan diagram for the integer Landau levels $N$ for different angles $\alpha$ of **B** in the $z$–$x$ plane (see inset; $\alpha$ is positive from the $z$ to the $x$ direction) as a function of **B**$^{-1}$. Data are obtained from the minima of $\rho_{xx}$ in extended data Fig. 4a. **i**, Shubnikov-de Haas frequency as a function of angle $\alpha$ and **j**. $\beta$. $\beta$ is the rotation angle of **B** in the $z$–$y$ plane (see inset; $\beta$ is positive from the $z$ to the $y$ direction). The black dots represent the measurement data. The red fitting curve represents a planar 2D Fermi surface, the blue fitting curve corresponds to an ellipsoidal 3D Fermi surface. **k** The Fermi surface of HfTe₅ in momentum space along the $k_x$, $k_y$, and $k_z$ direction.

Although those studies have provided signatures of field-induced states in the longitudinal electrical resistivity of Bi[24,25], ZrTe$_5$[21,26] and graphite[5,27–30], correlated states with Hall responses have yet to be observed.

In this work, we present measurements of the low-temperature longitudinal and Hall resistivities of the 3D semimetals HfTe$_5$ and ZrTe$_5$. Previous studies have shown that HfTe$_5$ is an iso-structural counterpart of ZrTe$_5$[31]. Both materials share an orthorhombic crystal structure and a single elliptical 3D Fermi surface, comprising less than 1% of the Brillouin zone and hosting massive Dirac Fermions with almost linearly dispersing bands in the vicinity of the Fermi level (see Supplementary Information for details). These specific properties have been considered essential for the observation of the 3D IQHE in both materials[21–23]. Moreover, recent progress in ZrTe$_5$[21] and HfTe$_5$[32] single-crystal growth has enabled a Hall mobility $\mu$ that exceeds 100,000 cm$^2$ V$^{-1}$ s$^{-1}$ at low temperatures (<4 K) (see "Methods" section and Supplementary Information S1). The quality of these crystals is comparable to that of graphene samples, which have previously proven appropriate for observing the FQHE in two dimensions[33]. While the 3D band structure of ZrTe$_5$ and HfTe$_5$ is very similar[21], hafnium has a higher atomic number than zirconium and hence naturally introduces stronger spin–orbit coupling (SOC)[31], which has been previously shown to stabilize correlated states in the quantum limit of 2D electron systems[34]. Therefore, HfTe$_5$ is the more promising candidate for the observation of unconventional Hall responses of correlated ground states in its quantum limit.

## Results

HfTe$_5$ and ZrTe$_5$ typically grow as millimeter-long ribbons with an aspect ratio of approximately 1:3:10, reflecting their crystalline anisotropy. Details of the growth process, crystal structure, and first transport characterization of our samples can be found in refs. [21,32]. We have measured the longitudinal electrical resistivity $\rho_{xx}$ and Hall resistivity $\rho_{xy}$ (see "Methods" section) of three HfTe$_5$ samples (A, B, C) and three ZrTe$_5$ samples (D, E, F) as a function of magnetic field $\mathbf{B}$ and temperature $T$, with the electrical current applied along the $a$-axis of the crystals.

At $T = 300$ K, $\rho_{xx}$ is around 0.5 m$\Omega$ cm (see Fig.1a, Supplementary Fig. S1 and ref. [32]) with an electron density of $n = 1.3 \times 10^{19}$ cm$^{-3}$ and $\mu = 10,000$ cm$^2$ V$^{-1}$ s$^{-1}$ [21,23,32]. Upon cooling in zero magnetic field, $\rho_{xx}$ increases with decreasing $T$ until it reaches a maximum at $T_L = 70$ K (Fig.1a). Such a maximum has previously been observed in HfTe$_5$[32] and ZrTe$_5$[21], and it is attributed to a Lifshitz transition, here inducing a change in charge-carrier type. Consistently, the slope of $\rho_{xy}(\mathbf{B})$ changes sign at $T_L$, indicating electron-type transport for $T < T_L$[32]. At 3 K, we find $n = 8.7 \times 10^{16}$ cm$^{-3}$ and $\mu = 250,000$ cm$^2$V$^{-1}$s$^{-1}$ (see Supplementary Note 1 and ref. [21]). All investigated samples show similar electrical transport properties. In the main text, we focus on data obtained from HfTe$_5$ sample A and ZrTe$_5$ sample D. Additional data of samples B, C, E, and F can be found in the Supplementary Information and ref. [23].

To characterize the Fermi-surface (FS) morphology of our pentatelluride samples, we have measured Shubnikov-de Haas (SdH) oscillations with respect to the main crystal axes at 3 K. For this purpose, we followed the analysis of ref. [21] and rotated $\mathbf{B}$ in the $z$–$y$ and $z$–$x$ planes, while measuring $\rho_{xx}$ ($\mathbf{B}$) at a series of different angles (Fig. 1, Supplementary Fig. S3 and ref. [23]). The SdH frequency $B_{F,i}$ is directly related to the extremal cross-section of the Fermi surface $S_{F,i}$, normal to the applied $\mathbf{B}$ direction via the Onsager relation $B_{F,i} = S_{F,i}(\hbar/2\pi e)$. Examples of the SdH oscillations for which the magnetic field was aligned along the three principal crystallographic directions ($x$, $y$, and $z$ axes) are shown

in Fig. 1b–d (upper panels). In each field direction, we have observed maxima in $\rho_{xx}$ that are periodic in $1/\mathbf{B}$, each of which corresponds to the onset of a Landau level. In the associated minima, $\rho_{xx}(\mathbf{B})$ does not vanish, which is a consequence of the remaining dispersion in $z$ direction in 3D systems and Landau level-broadening due to disorder (see Supplementary Note 4, Supplementary Fig. S4–S7 and ref. [23] for details). To determine the SdH oscillation frequency, we have subtracted the smooth high-temperature (50 K)—$\rho_{xx}(\mathbf{B})$ from the low-$T$-data, obtaining the oscillating part of the longitudinal resistivity $\Delta\rho_{xx}(\mathbf{B})$. Employing a standard Landau-index fan diagram analysis to $\Delta\rho_{xx}(\mathbf{B})$ (Fig. 1h, Supplementary Figs. S3, S8, S9, Supplementary Note 2 and ref. [23]), we have found only a single oscillation frequency for all rotation angles measured, consistent with a single electron pocket at the Fermi energy. The extracted $B_{F,i}$ of sample A for $\mathbf{B}$ along the three principal directions are $B_{F,x} = (9.9 \pm 0.1)$ T, $B_{F,y} = (14.5 \pm 0.5)$ T, and $B_{F,z} = (1.3 \pm 0.1)$ T. Here, the errors denote the standard deviation of the corresponding fits.

In contrast to 2D materials, HfTe$_5$ and ZrTe$_5$ show in-plane SdH oscillations when $\mathbf{B}$ is aligned with $x$ and $y$, indicating a 3D Fermi-surface pocket. The shape of the FS is further determined by the analysis of the rotation angle-dependence of $B_F$. As shown in Fig. 1i, j, the angle-dependent SdH frequency is well represented by a 3D ellipsoidal equation $B_{F,3D} = B_{F,z}B_{F,i}/\sqrt{(B_{F,z}\sin\theta)^2 + (B_{F,i}\cos\theta)^2}$, where $\theta$ is the rotation angle in the $z$–$i$ plane. As a cross-check, we also plot the formula of a 2D cylindrical Fermi surface $B_{F,2D} = B_{F,z}/\cos\theta$, which deviates significantly from the experimental data for $\theta > 80°$. Hence, the ellipsoid equations can be used to obtain the Fermi wave vectors $k_{F,x} = \sqrt{S_{F,y}S_{F,z}}/\sqrt{\pi S_{F,x}} = (0.005 \pm 0.001)$ Å$^{-1}$, $k_{F,y} = \sqrt{S_{F,x}S_{F,z}}/\sqrt{\pi S_{F,y}} = (0.008 \pm 0.001)$ Å$^{-1}$ and $k_{F,z} = \sqrt{S_{F,x}S_{F,y}}/\sqrt{\pi S_{F,z}} = (0.058 \pm 0.006)$ Å$^{-1}$ that span the 3D FS of HfTe$_5$ sample A in the $x$, $y$, and $z$ direction, respectively (Fig. 1k). The errors in $k_{F,i}$ originate from the errors of the $B_{F,i}$. The preceding analysis indicates that for our HfTe$_5$ and ZrTe$_5$ samples, the quantum limit with the field along the $z$ is achieved already for the field of $B_C = 1.8$ and 1.2 T[21,23], respectively. Further details of our band-structure analysis can be found in Supplementary Fig. S6, Supplementary Table S1, refs. [21,23]. Above 6 T, we find that for both materials, $\rho_{xx}(\mathbf{B})$ steeply increases with the magnetic field (Fig. 2b). Such a steep increase has previously been observed in ZrTe$_5$ and has been attributed to a field-induced metal-insulator transition[21].

For the field-aligned with the $z$ axis ($\mathbf{B}||z$), we additionally observed in both studied compounds pronounced plateaus in Hall resistance $\rho_{xy}(\mathbf{B})$ that appear at the minima of the SdH oscillations in $\rho_{xx}(\mathbf{B})$—features commonly related to the QHE (Fig. 2a and ref. [23]). The height of the last integer plateau is given by $(h/e^2) \pi/k_{F,z}$, similar to as reported in the literature for the 3D IQHE[21,22]. The plateaus are most pronounced at low temperatures, but still visible up to $T = 30$ K (Fig. 2b, c and ref. [23]).

We note that the observed quantization of $\rho_{xy}$ is not immediately obvious from the predicted quantization in $\sigma_{xy}$. The Hall resistivity tensor is given by $\rho_{xy} = \sigma_{xy}/(\sigma_{xx}\sigma_{yy} + \sigma_{xy}^2)$ with a magnetic field in $z$ direction, where $\sigma_{xx}$ and $\sigma_{xx}$ are the longitudinal component of the conductivity tensor in $x$ and $y$ direction, respectively. Vice versa, the Hall conductivity tensor element is given by $\sigma_{xy} = \rho_{xy}/(\rho_{xx}\rho_{yy} + \rho_{xy}^2)$ with a magnetic field in $z$ direction. However, in our samples at low temperatures $\sigma_{xx} < \sigma_{xy}$ (Supplementary Figs. S11 and S12) and thus $\sigma_{xy}^{-1} \approx \rho_{xy}$, enabling

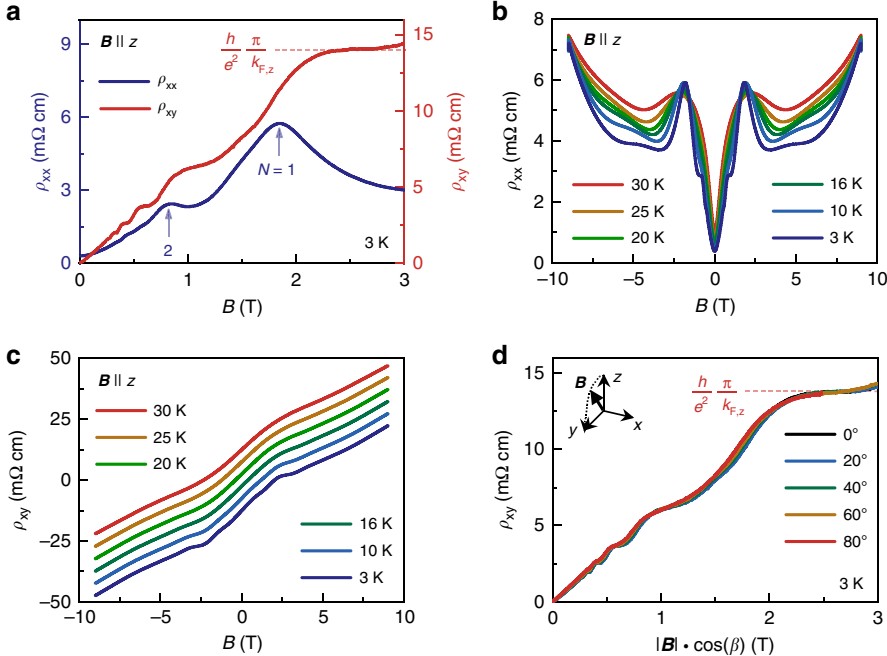

**Fig. 2 Three-dimensional integer quantum Hall effect in HfTe$_5$. a** Longitudinal electrical resistivity $\rho_{xx}$ (blue, left axis) and Hall resistivity $\rho_{xy}$ (red, right axis) and as a function of **B** at $T = 3$ K with **B** applied in $z$. The blue arrows mark the onset of a Landau level (LL). The blue numbers denote the index $N$ of the corresponding LL. The plateau in $\rho_{xy}$ scales with $(h/e^2)\,\pi/k_{F,z}$, with the Planck constant $h$, the electron charge $e$, and the Fermi wavevector in $z$ direction $k_{F,z}$. **b** $\rho_{xx}$ and **c** $\rho_{xy}$ as a function of **B** for various temperatures $T \geq 3$ K with **B** applied in $z$. **d** $\rho_{xy}$ as a function of $|\mathbf{B}|\cos(\beta)$ for magnetic fields along the direction with angles $\beta$ at 3 K.

the direct observation of the quantization. Due to the geometry of the HfTe$_5$ crystals (elongated needles) and their mechanical fragility, performing reliable measurements of $\rho_{yy}$ is not possible. Instead, we estimate the error of the $\sigma_{xy}$ using the ratio of Drude resistivities $\rho_{yy}/\rho_{xx} = (n_{xy}e^2 t_x/m_x^*)/(n_{xy}e^2 t_y/m_y^*)$ given by the quantum lifetimes and effective masses obtained from Shubnikov-de Haas oscillations on sample A (Supplementary Table S1). $n_{xy}$ is the charge-carrier concentration in the $x$–$y$ plane. Based on this analysis, we find $\rho_{yy}/\rho_{xx} \approx 0.4$, which results in an error of below 8 % in the estimated $\sigma_{xy}$ for the investigated field range owing to $\rho_{xx}(B) < \rho_{xy}(B)$. Both these errors lay within the estimated error of $k_{F,z}$ of 10 %.

Figure 2d shows the angular-dependence of the Hall plateaus, which we find to scale with the rotation angle. This behavior is very similar to the sister compound ZrTe$_5$.[21] In both materials, the height of the Hall plateaus and its position in $|\mathbf{B}|$ depends only on the field component that is perpendicular to the $x$–$y$ plane $B_\perp = |\mathbf{B}|\cos\theta$[21,23].

So far, our analysis focused on similarities between the Hall effects observed in ZrTe5 and HfTe5. Upon cooling the samples to 50 mK, an obvious difference emerges, as shown in Fig. 3 and Supplementary Figs. S11–S15. At low fields below the quantum limit ($B < B_C$), both compounds exhibit signatures of new peaks and plateaus in $\rho_{xx}(B)$ and $\rho_{xy}(B)$. Such features have been observed in the past and are related to spin splitting of the Landau levels[23,35]. However, at high fields ($B > B_C$)—in the quantum limit, HfTe5 exhibits an additional peak in $\rho_{xx}(B)$, accompanied by a plateau-like feature in $\rho_{xy}(B)$. This is in sharp contrast to ZrTe5, in which $\rho_{xx}(B)$ and $\rho_{xy}(B)$ smoothly increase. Using the Landau-index fan diagram obtained at 3 K and gauging the indexing of Landau bands with respect to the $N = 1$ band, we find that the additional peak in $\rho_{xx}(B)$, in the quantum limit of HfTe5 is situated at $N = 3/5$. This indexing is confirmed by corresponding maxima in $\rho_{xx}(B)$ and/or $\sigma_{xx}(B)$ of all three HfTe5

samples investigated (compare Supplementary Figs. S11 and S12), despite being less pronounced in some of them.

Although the relation between the magnitude of the plateau-like feature and its corresponding Landau index is not obvious from $\rho_{xy}(B)$, a comparison of the respectively calculated conductivity reveals that the magnitude of the plateau-like feature in the quantum limit scales as 3/5-times with respect to the plateau related to the LLL. A close investigation of the plateau height in conductivity (Supplementary Fig. S16) reveals that both the plateau-like features at $N = 1$ and $N = 3/5$ are well developed in the conductivity, within 1 and 2% of $N \cdot (e^2/h)k_{F,z}/\pi$ in the range of 0.5 T around the plateau center.

In order to verify whether the observed features can be explained by invoking the presence of a second pocket at the Fermi energy, we have performed additional magneto-transport measurements up to 70 T (Supplementary Fig. S16 and ref. [23]). The measurements did not reveal any additional quantum oscillations, which is consistent with band-structure calculations[31] and a previous ARPES study on our samples[36]: The Fermi level, obtained from the analysis of the Shubnikov-de Haas oscillations is $(9 \pm 2)$ meV (Supplementary Information), which is in agreement with the ARPES experiment. According to the ARPES data, at 15 K, the nearest additional band is located ~5 meV above the Fermi level (lowest temperature measured in the ARPES study) as compared to the Fermi function broadening of $k_B \cdot 15$ K $\approx 1$ meV. Below 15 K, the Fermi level stays constant with respect to the band edges, as indicated by the temperature-independent Shubnikov-de Haas frequency in our experiments. Hence, the next nearest band in our samples is ~$k_B \cdot 60$ K away from the Fermi level and does not contribute to the low-temperature transport experiments. Our data can, therefore, be analyzed in terms of a single electron-type Dirac pocket.

Further insight into the possible origin of the $N = 3/5$ state in the quantum limit can be obtained from the line shape of $\Delta\rho_{xx}(B)$,

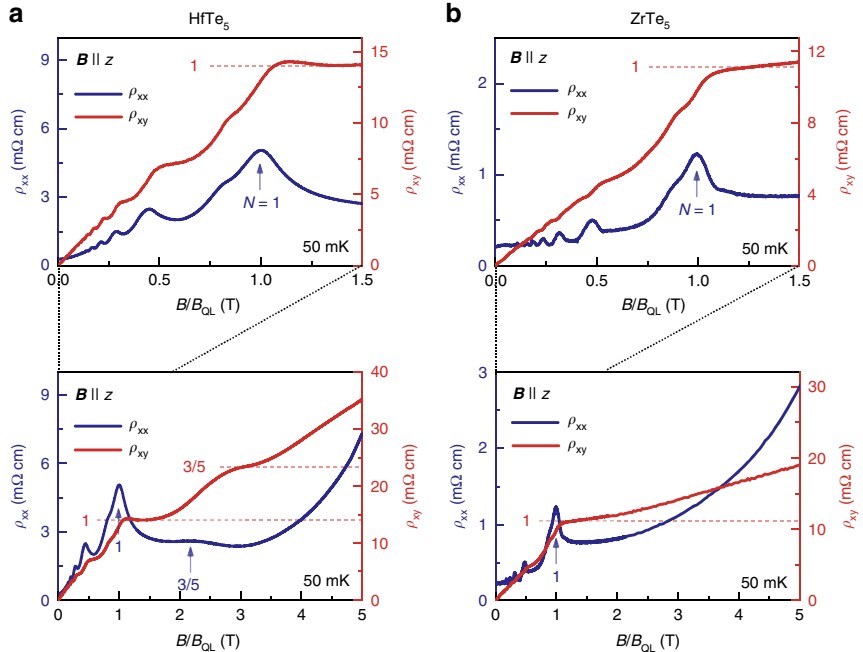

**Fig. 3 Low-temperature longitudinal magnetoresistivity and Hall resistivity in isostructural HfTe₅ and ZrTe₅ at 50 mK. a** Longitudinal electrical resistivity $\rho_{xx}$ (blue, left axis) and Hall resistivity $\rho_{xy}$ (red, right axis) of ZrTe₅ as a function of $\mathbf{B}/B_{QL}$ at $T = 50$ mK with the magnetic field **B** applied in $z$ for $0 \text{ T} \leq \mathbf{B} \leq 3 \text{ T}$ (upper panel) and $0 \text{ T} \leq \mathbf{B} \leq 9 \text{ T}$ (lower panel). The blue arrows mark the onset of the Landau levels. $\mathbf{B_{QL}}$ denotes the magnetic field of the onset of the $N = 1$ Landau level. The blue numbers label the index $N$ of the Landau level and the red numbers label the corresponding value of $\rho_{xy}$ with respect to $(h/e^2)\pi/k_{F,z}$. **b** Longitudinal electrical resistivity $\rho_{xx}$ (blue, left axis) and Hall conductivity $\sigma_{xy}$ (red, right axis) of ZrTe₅ as a function of **B** at $T = 50$ mK with **B** applied in $z$ for $0 \text{ T} \leq \mathbf{B} \leq 3 \text{ T}$ (upper panel) and $0 \text{ T} \leq \mathbf{B} \leq 9 \text{ T}$ (lower panel).

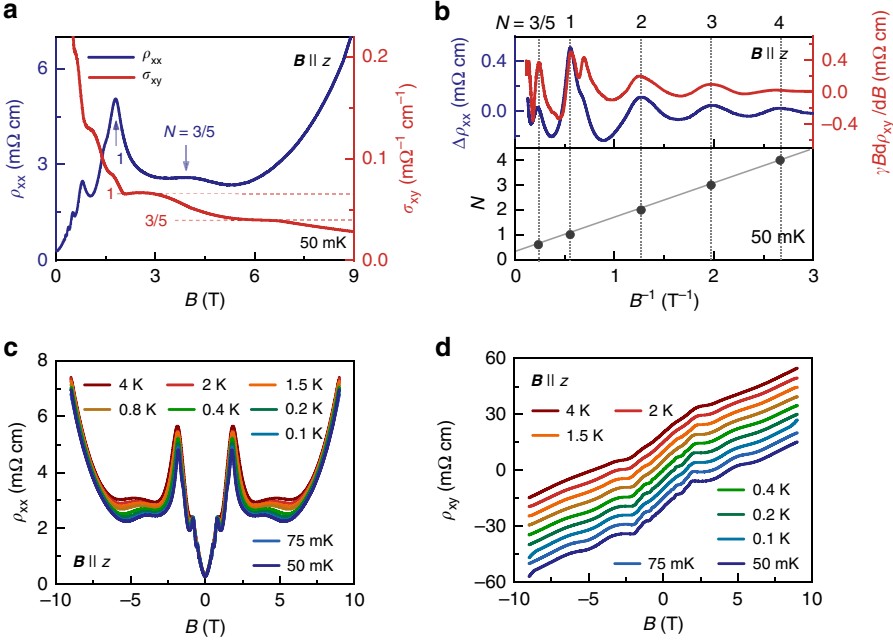

**Fig. 4 Three-dimensional Hall response in the quantum limit of HfTe₅. a** Longitudinal electrical resistivity $\rho_{xx}$ (blue, left axis) and Hall resistivity $\rho_{xy}$ (red, right axis) and as a function of **B** at $T = 50$ mK with **B** applied in $z$ for $0 \text{ T} \leq \mathbf{B} \leq 3 \text{ T}$ (upper panel) and $0 \text{ T} \leq \mathbf{B} \leq 9 \text{ T}$ (lower panel). The blue arrows mark the onset of a Landau level. The blue numbers label the index $N$ of the Landau level and the red numbers label the corresponding value of $\sigma_{xy}$ with respect to $(e^2/h)k_{F,z}/\pi$. **b** Variation of the longitudinal electrical resistivity $\Delta\rho_{xx}$ as a function of **B** (upper panel, left axis), $\gamma \mathbf{B} \cdot d\rho_{xy}/d\mathbf{B}$ (right axis, upper panel) ($\gamma = 0.04$) and Landau-index fan diagram (lower panel) as a function of $\mathbf{B}^{-1}$ at $T = 50$ mK with **B** applied in $z$. **c** $\rho_{xx}$ and **d** $\rho_{xy}$ as a function of **B** for various temperatures $4 \text{ K} \geq T \geq 50$ mK with **B** applied in $z$.

which resembles the line shape of $\sigma_{xx}(B)$, (Fig. 4e, f) a common feature of canonical 2D QHE systems[37] A related empirical observation is that in both fractional and IQHE in 2D systems the longitudinal resistance is $\rho_{xx}(\mathbf{B})$ is connected to $\rho_{xy}(\mathbf{B})$ via $\rho_{xx}(B_z) = \gamma B \cdot d\rho_{xy}(\mathbf{B})/d\mathbf{B}$, where $\gamma$ is a dimensionless parameter of the order of 0.01–0.05, which measures the local electron concentration fluctuations[38,39]. Comparison of $\sigma_{xx}(\mathbf{B})$ (Fig. 4d) and $\gamma B \cdot d\rho_{xy}(\mathbf{B})/d\mathbf{B}$ (Fig. 4e, upper panel) as a function of $\mathbf{B}^{-1}$ reveals that both quantities show maxima and minima at the same field positions as $\Delta\rho_{xx}(\mathbf{B})$. In particular, the derivative relation is well fulfilled with $\gamma = 0.04$, which is in the expected range reported for 2DESs. These results suggest that the observed plateau-like feature observed in the quantum limit in HfTe$_5$ is related to quantum Hall physics.

To gain quantitative insights into the states that cause the features in the Hall effect in HfTe$_5$, we have estimated the gap energies of the $x$–$y$ plane $\Delta_N$ associated with $N = 1$ and $N = 2$ below and $N = 3/5$ in the quantum limit. We have fitted the $T$-dependence of the $\rho_{xx}(B)$ minima (Fig. 4f) in the thermally activated regime $\rho_{xx}(\mathbf{B}) \propto \exp(-\Delta_N/2k_BT)$, where $k_B$ is the Boltzmann constant (Supplementary Fig. S17). For integer $N$, we find $\Delta_1 = (40 \pm 2)$ K at $N = 1$ and $\Delta_2 = (9 \pm 1)$ K at $N = 2$. The gap energy of the $N = 3/5$ state in the quantum limit is two orders of magnitude lower: $\Delta_{3/5} = (0.49 \pm 0.09)$ K. The deviations given for the gaps are the errors obtained from the thermally activated fits in Supplementary Fig. S13. In spite of considerable Landau level-broadening, both the size of the gaps of the integer and the $N = 3/5$ states compare well with the gaps obtained for integer and correlated quantum Hall states in 2DESs[37,40]. The different $\Delta_N$ are also in agreement with the $T$-dependence of the corresponding Hall features (Fig. 4g). While the integer plateaus are observable up to tens of Kelvin, the plateau-like feature in the quantum limit vanishes at around 0.5 K.

Those considerations suggest that the Hall feature observed in the quantum limit of HfTe$_5$ is associated with physics in the LLL only (as pointed out above, this Landau level is non-degenerate since HfTe$_5$ is like ZrTe$_5$ a gapped Dirac semimetal). The finite value of $\rho_{xx}$ at $N = 3/5$ implies the absence of a fully established bulk gap, which in turn means that a truly quantized Hall effect as in 2D systems without the $k_{F,z}$-scaling cannot be expected. Nevertheless, the emergence of the plateau-like feature in the quantum limit of a single-band system at low temperatures calls for an explanation beyond a simple single-particle picture: The Hall conductivity of a non-interacting single band in which the chemical potential adjusts to keep the particle number fixed simply decreases as $1/\mathbf{B}$, as observed in the isostructural ZrTe$_5$ (Fig.3b).

## Discussion

Although possible scenarios for the emergence of a plateau in the Hall resistance in the quantum limit of electron plasma include the formation of a CDW, Luttinger liquid, Wigner crystallization, or the so-called Hall crystal[41,42], a favorable scenario builds on the notion that ZrTe$_5$ and HfTe$_5$ can be thought of as a stack of interacting 2DEGs. Based on a Hartree–Fock analysis, it was proposed[13] that in a layered structure the gain in exchange energy can exceed the energy cost for distributing electrons unequally between layers. The electrons then undergo spontaneous staging transitions in which only every $i$-th layers is occupied, while all other layers are emptied (the number $i$ depends on the average electron density and the state formed)—some of which are only stabilized due to the interplay of electron interaction and spin–orbit coupling[43]. Depending on layer separation, electron density, and the strength of electron–electron interactions,

various types of layered Laughlin states or Halperin states can then be formed[3,13,14]. These states are naturally associated with Hall responses. While staging transitions are unlikely in isotropic three-dimensional materials at high electron densities, HfTe$_5$ has a very anisotropic band structure with small tunneling amplitudes along $z$, and hosts only a relatively small number of electrons in its Dirac pocket. Our data are thus consistent with strong interactions stabilizing a correlated state that gives a Hall response in HfTe$_5$ in the quantum limit.

In conclusion, our measurements reveal an unconventional correlated electron state manifested in the Hall conductivity of the bulk semimetal HfTe$_5$ in the quantum limit, adjacent to the 3D IQHE at lower magnetic fields. The observed plateau-like feature is accompanied by a Shubnikov-de Haas minimum in the longitudinal electrical resistivity and its magnitude is approximately given by $3/5(e^2/h)k_{F,z}/\pi$. Analysis of derivative relations and estimation of the gap energies suggest that this feature is related to quantum Hall physics. The absence of this unconventional feature in the quantum limit of isostructural single-band ZrTe$_5$ samples with similar electron mobility and Fermi wavevector indicates the presence of a correlated state that may be stabilized by spin–orbit coupling. However, further experimental and theoretical efforts in determining the real interactions and texture of the field-induced correlated states in HfTe$_5$ are necessary to settle the puzzle of the unconventional Hall response in the quantum limit. In particular, experiments directly probing the density of states and the real space charge distribution such as Scanning Tunneling Spectroscopy and in-field X-ray diffraction could shed additional light on the nature of the observed feature.

## Methods

**Single-crystal sample growth and pre-characterization**. Single crystals of HfTe5 were obtained via a chemical vapor transport method. Stoichiometric amounts of Hf (powder, 3 N) and Te (powder, 5 N) were sealed in a quartz ampoule with iodine (7 mg ml$^{-1}$) and placed in a two-zone furnace. A temperature gradient in the range of 400–500 °C was applied. After ca. 1-month, long ribbon-shaped HfTe5 single crystals were extracted from the ampule with a typical size of the single crystals 1 mm × 0.5 mm × 3 mm (width × height × length). High-quality needle-shaped (about 0.1 × 0.3 × 20 mm$^3$) single crystals of ZrTe5 were synthesized using the tellurium flux method and high-purity elements (99.9999% zirconium and 99.9999% tellurium). The lattice parameters of the crystals were confirmed by single-crystal X-ray diffraction. The samples used in this work are of the same batch as the ones reported in refs. [21,23,32,36] and have similar Fermi level positions. As shown in these papers, in our HfTe5 and ZrTe5 samples a three-dimensional topological Dirac semimetal state emerges only at around $T_p \approx 65$ K (at which the resistivity shows a pronounced peak), manifested by a large negative magnetoresistance. This Dirac semimetal is a critical point between two distinct topological insulator phases: weak ($T > T_p$) and strong ($T < T_p$). At high temperatures, the extracted band gap is around 30 meV (185 K), and at low temperatures 10 meV (15 K)[36]. However, we note that the Fermi level at these temperatures is not located in the gap, but several meV in the valence band for $T > T_p$ and in the conduction band for $T < T_p$. Hence, our HfTe5 and ZrTe5 samples are metallic at both high and low temperatures.

**Electrical transport measurements**. Electrical contacts to the HfTe5 and ZrTe5 single crystals were defined with an Al hard mask. Ar sputter etching was performed to clean the sample surface prior to the sputter deposition of Ti (20 nm) and Pt (200 nm) with a BESTEC UHV sputtering system. Subsequently, Pt wires were glued to the sputtered pads using silver epoxy. All electrical transport measurements up to ±9 T were performed in a temperature-variable cryostat (PPMS Dynacool, Quantum Design), equipped with a dilution refrigerator inset. To avoid contact-resistance effects, only four-terminal measurements were carried out. The longitudinal $\rho_{xx}$ and Hall resistivity $\rho_{xy}$ were measured in a Hall-bar geometry with standard lock-in technique (Zurich instruments MFLI and Stanford Research SR 830), applying a current of 10 μA with a frequency of $f = 1$ kHz across a 100 kΩ shunt resistor. The electrical current is always applied along the $a$-axis of the crystal.

The pulsed magnetic field experiments up to 70 T were carried out at the Dresden High Magnetic Field Laboratory (HLD) at HZDR, a member of the European Magnetic Field Laboratory (EMFL).

                                                                ARTICLE

## Data availability

All data generated or analyzed during this study are available within the paper and its Supplementary Information file. Reasonable requests for further source data should be addressed to the corresponding author.

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

## Acknowledgements

We thank Andrei Bernevig for fruitful discussions. T.M. acknowledges financial support by the Deutsche Forschungsgemeinschaft via the Emmy Noether Program ME4844/1-1 (project id 327807255), the Collaborative Research Center SFB 1143 (project id 247310070), and the Cluster of Excellence on Complexity and Topology in Quantum Matter ct.qmat (EXC 2147, project id 390858490). C.F. acknowledges the research grant DFG-RSF (NI616 22/1): Contribution of topological states to the thermoelectric properties of Weyl semimetals and SFB 1143. G.F.C. was supported by the Ministry of Science and Technology of China through Grant No. 2016YFA0300604, and the National Natural Science Foundation of China through Grant No. 11874417. T.F. and J.W. acknowledge support from the DFG through the Würzburg-Dresden Cluster of Excellence on Complexity and Topology in Quantum Matter ct:qmat (EXC 2147, project-id 39085490), the ANR-DFG grant Fermi-NESt, and by Hochfeld-Magnetlabor Dresden (HLD) at HZDR, member of the European Magnetic Field Laboratory (EMFL). P.M.L., Q.L., and G.G. acknowledge support from the Office of Basic Energy Sciences, U.S. Department of Energy (DOE) under Contract No. DE-SC0012704. J.G. acknowledges support from the European Union's Horizon 2020 research and innovation program under Grant Agreement ID 829044 "SCHINES".

## Author contributions

S.G. and J.G. conceived the experiment. X.Z., W.Z., and G.F.C. synthesized and pre-characterized the single-crystal HfTe₅ bulk samples. P.M.L., Q.L., and G.G. synthesized and pre-characterized the single-crystal ZrTe₅ bulk samples. S.G. and S.H. fabricated electrical transport devices. A.M. and C.F. sputtered the electrical contacts on the samples. S.G. carried out the low-field transport measurements with the help of S.H., J.G., and R.W. S.G., J.G., T.F., and J.W. carried out the high-field transport experiments. T.E. and T.M. provided the model of the three-dimensional quantum Hall effect. S.G., N.L., T.M., T.F., and J.G. analyzed the data. All authors contributed to the interpretation of the data and to the writing of the manuscript.

## Funding

## Competing interests

The authors declare no competing interests.

**Additional information**

