## [Peer Review File · Nature Communications]

Reviewers' comments:

Reviewer #1 (Remarks to the Author):

In this work, Galeski et al. report transport properties of measurement in HfTe5 bulk crystals. As the main message, in the quantum limit regime, they discovered a new "signature" of Hall plateaus are accompanied by Shubnikov-de Hass oscillation minima in extra-low temperatures. In this result, the authors explained it as a fractional quantum Hall response with strong electron-electron interactions. Regarding the novelty criteria, this work seems to be the first observation of FQH response in the 3D HfTe5 system. I would like to recommend this manuscript to be accepted in Nature communications if they can clarify my following concerns:

(1) My first concern is that the authors explain the peak deriving from the first Landau level $\nu=1$ as "fractional $6/5$ ". In figure 3 (a)&(b), there is little feature between $\nu=1$ and $\nu=2$. However, it is not well-developed peak or plateaus. It worth noting similar materials ZrTe5 and results in the previous published papers [ref.22, Yanwen Liu et. al, Nature Communications 7,12516 (2016), Zheng G. et al, PRB 93, 115414 (2016)], those adjacent peaks are interpreted as the Zeeman splitting mechanism. Thus, there is possibly alternative understanding by the spin-splitting as spin-up and spin-down electrons. Can authors exclude or distinguish the "fractional state $6/5$ " from "Zeeman splitting of spin up and down"?

(2) The authors believe their observation of "3D FQH" response in the extreme quantum limit regime. However, unlike the FQHE in a 2D electronic system, this observation is only appearing the "fractional $6/5$ and $3/5$ " instead of the robust fractional " $1/3$ " state in usual 2D system. To explain the 3D fractional composite fermions, it is essentially required to form a full energy gap with strong MANY-BODY interactions. However, I notice that the minima of $\rho_{xx}(B)$ is still finite (such as at the position of $\nu=6/5, 1, 3/5, \sim 3\text{mOhms.cm}$, in Figure 3 (a)), and the authors interpreted it as 3D Hall effect with disorder in SI(S4). Those explanations are very strange to me, because the plateaus $\nu=1$ is attributed to the disorder (single particle picture, if I understand it correctly), whereas its front ($\nu=6/5$) and rear ($\nu=3/5$) regimes correspond to strong MANY-BODY interactions. This certainly requires further clarification, especially in the magnetic field direction.

(3) A comment in page 6 (line 139), the Hall resistivity tensor should be written as: $\rho_{xy} = \sigma_{xy} / (\sigma_{xx} \sigma_{yy} + \sigma_{xy}^2)$, and a short description of approximation needs to be added. Otherwise, it is quite misleading in an anisotropic system (since $\sigma_{xx}(B) \neq \sigma_{yy}(B)$).

(4) Page 5, line 103, there is a typo in pdf.

In summary, the authors had studied the magneto-transport properties of Dirac semimetal HfTe5, and they convincingly present a pure 3D version quantum Hall effect (3D QHE) in another system. In addition, at ultra-quantum limit region, they discovered an interesting "fractional" state in 3D system. Although it still needs fresher experimental and theoretical attentions, those results may open a research avenue in the subject of field-induced many-body effects in 3D electronic system.

Reviewer #2 (Remarks to the Author):

The manuscript reported the observation of fractional plateaus in HfTe5. The three-dimensional integer quantum Hall effect in ZrTe5 has brought in wide interest for stacked layer materials. The authors creatively used HfTe5, with a structure similar to ZrTe5, but hafnium is heavier than zirconium so stronger spin-orbit coupling is expected. The sample used in this study is also of high quality and features of $\nu=1$ integer quantum Hall effect can be observed. However, the data did not appear to support the existence of fractional quantum Hall effect at $6/5$ and $5/3$.

In line 133 and 134, the authors stated that “pronounced plateaus in Hall resistance.....features commonly related to the QHE (Fig 2a)”, but none of the plateaus in Fig. 2a is pronounced. Only the $\nu=1$ shoulder is plateau-like. The readers deserved to know how flat it is and the authors should provide a zoom-in. The Hall measurement in three-dimensional material is challenging, and a barely ideal plateau may also support the existence of $\nu=1$ integer quantum Hall state. Only based on the assumption of $\nu=1$ shoulder, one can guess the existence of $\nu=2$ state. Otherwise, a slope change in Hall resistance and an oscillation in longitudinal resistance around 1T are not as clear as that around 2.5T.

Then, if we focus on Fig. 3a, the slope changes labelled as $6/5$ and $3/5$ don't provide sufficient evidence to be fractional quantum Hall plateaus. There are other possible causes of the slope change in Hall resistance, and fractional quantum Hall effect is not the most straight forward explanation. Although I appreciate the efforts the authors have spent on this experiment, I still want to persuade the authors don't claim the fractional quantum response with the existing data. The observation of only the $\nu=1$ three-dimensional integer quantum Hall effect in HfTe5 is probably a better conclusion.

Last but not least, I have a technical concern. The longitudinal resistance corresponding to the expected plateaus should be zero or a dip. The authors intentionally labelled the filling factor at the peak position, which is much easier to see. Traditionally, the filling factor is labelled at the magnetic field of quantum Hall plateaus. A dip changing with temperature or a zero longitudinal resistance at the filling factor of plateau is much stronger evidence than a peak next to a plateau. At filling factors of $\nu=1$ and 2, there are dips, and the authors may check the temperature dependence to estimate the energy gap. At filling factors of $\nu=6/5$, there is no minimum in longitudinal resistance in Fig. 3a.

Reviewer #3 (Remarks to the Author):

S. Galeski¹ et al. have studied the quantum oscillations of HfTe5 and observed Hall-plateau-like features at both integer and fractional fillings. Based on the value of the plateaus, they have confirmed the 3D integer quantum Hall effect, previously observed in a similar material ZrTe5. They claim that these plateaus at fractional fillings are fractional quantum Hall effect. 3D quantum integer Hall effect is a recent development in the quantum Hall effect field. It is absolutely intriguing if the fractional version of the effect can be realized. Such studies are very welcome. Although similar claims on a similar material have been made before, as in Ref. 19, 22, determination of the fractional filling factor in these earlier studies were only based on the resistivity oscillation. This work demonstrates quantized Hall plateaus and represents a progress in this field. However, there are inconsistencies that undermine the reliability of the claim. Unless these inconsistencies are resolved, I cannot recommend its publication in Nature Comm.

In the 2D quantum Hall effect, determination of the filling factor can be done with little uncertainty/ambiguity, because the quantized Hall plateau is in unit of resistance and only depends on physical constants. However, it is not the case for the 3D quantum Hall effect, where estimation of the value of the plateau relies on accurate measurement of the sample geometry and the kF . As for the geometry, it is easy to have an error of 10% or more, not to mention the error in kF . This problem becomes more severe for fractional fillings. Therefore, it is of paramount importance to make sure that the determined filling factors are accurate and reliable. Regarding this inherent difficulty, I think that the following concerns should be addressed.

The authors estimate the filling factor from the value of the Hall plateau, which they claim is confirmed

by the resistance oscillations. According to Fig. 3a and 3e, the slope of the Hall is substantially reduced above the $n=1$ Landau level. This suggests that to arrive at the next plateau, for instance, $\nu=3/5$, one needs to go to a field higher than expected from the low field slope. Correspondingly, the oscillation period of the resistance will be expanded. So, the filling factors obtained by two methods should differ rather than agree.

For a Dirac dispersion, one anticipates a half-integer quantum Hall effect due to the non-trivial Berry phase, as seen in graphene. Taking into account the spin degeneracy, it is expected to see plateaus at filling factors of 1, 3, 5,... When Zeeman splitting is strong enough, even filling factors start to develop. For a parabolic dispersion, even filling factors should appear, that is, 2, 4, 6,... Odd filling factors will be produced by the Zeeman splitting at high fields. In this experiment, integer filling factors are equally spaced. One possibility is that the spin degeneracy is lifted for all observed Landau levels, even those occurring at low fields. This may not be easily justified. Note that previous study on ZrTe₅ have shown that Zeeman splitting of Landau levels appears only at high fields, Nat. Commun. 7, 12516 (2016). The second possibility is that the filling factors are in fact 2, 4, 6...instead of 1, 2, 3... Consequently, the fractional filling factors observed in this study could be due to the Zeeman splitting of $n=2$ and 4 Landau levels. Comparing the Hall slope and the SdH oscillation frequency may be able to tell whether the spin degeneracy is lifted.

A minor comment. In Fig. 3, all the arrows, red or blue, seem to represent the Landau level index n (the field at which a Landau level is aligned with the Fermi level) rather than the filling factor ν . It is confusing when these two are used interchangeably.

Response to the Reviewers

Reviewer #1

Dear Reviewer,

thank you for the elaborate and detailed review. We are sure it will help us improve the quality of the manuscript. We find your appreciation of our work very gratifying and that you believe our results to be a possible path to 'open a research avenue in the subject of field-induced many-body effects in 3D electronic systems'. In the following, we would like to address the points that were raised in the review in the same sequence:

1. *My first concern is that the authors explain the peak deriving from the first Landau level $\nu=1$ as "fractional 6/5". In figure 3 (a)&(b), there is little feature between $\nu=1$ and $\nu=2$. However, it is not well-developed peak or plateaus. It worth noting similar materials ZrTe₅ and results in the previous published papers [ref.22, Yanwen Liu et. al, Nature Communications 7,12516 (2016), Zheng G. et al, PRB 93, 115414 (2016)], those adjacent peaks are interpreted as the Zeeman splitting mechanism. Thus, there is possibly alternative understanding by the spin-splitting as spin-up and spin-down electrons. Can authors exclude or distinguish the "fractional state 6/5" from "Zeeman splitting of spin up and down"?*

This is a point that we also found concerning. It is now clear that the possibility of the 6/5 state originating from spin splitting has not been addressed in sufficient detail in the previous version of the manuscript. In the current version we have, therefore, included a 1:1 comparison between the low temperature Hall and longitudinal resistivity of ZrTe₅ and HfTe₅ (Figure 3 of the revised manuscript). The comparison has revealed that at low fields, below the quantum limit, both materials show the same behavior rendering the feature originally labeled at 6/5 most probably a signature of spin splitting. However, significant differences appear in the quantum limit. For ZrTe₅, the Hall resistivity is as expected a smooth function of $1/B$, whereas for HfTe₅ a plateau-like feature accompanied by a maximum in longitudinal resistivity is present at the 3/5 filling. Owing to the structural an electronic similarity between ZrTe₅ and HfTe₅, the robust high field feature in HfTe₅ has to be considered to emerge from a many-body effect stabilized by the increased spin-orbit coupling in HfTe₅. Below we list the changes made in the manuscript in order to address this issue:

- *"So far, our analysis focused on similarities between the IQHEs observed in ZrTe₅ and HfTe₅. Upon cooling the samples to 50 mK, an obvious difference emerges, as shown in Fig. 4a investigated (Supplementary Fig. S11 - S14). At low fields below the quantum limit ($B < B_C$), both compounds exhibit signatures of new peaks and plateaus in $\rho_{xx}(B)$ and $\rho_{xy}(B)$. However, such features have been observed in the past and are related to spin splitting of the Landau levels.^{21,29}"*

- “Fig. 3. Low temperature longitudinal magnetoresistivity and Hall resistivity in isostructural HfTe_5 and ZrTe_5 at 50 mK. **a**, Longitudinal electrical resistivity ρ_{xx} (blue, left axis) and Hall resistivity ρ_{xy} (red, right axis) of ZrTe_5 as a function of $\mathbf{B}/\mathbf{B}_{OL}$ at $T = 50$ mK with the magnetic field \mathbf{B} applied in z for $0 T \leq \mathbf{B} \leq 3 T$ (upper panel) and $0 T \leq \mathbf{B} \leq 9 T$ (lower panel). The blue arrows mark $\nu = 3/5$ and $\nu = 1$ filling factors. \mathbf{B}_{OL} denotes the magnetic field of the onset of the $\nu = 1$ Landau level. The blue and red numbers label the filling factor ν of the corresponding Landau level. **b**, Longitudinal electrical resistivity ρ_{xx} (blue, left axis) and Hall conductivity σ_{xy} (red, right axis) of ZrTe_5 as a function of \mathbf{B} at $T = 50$ mK with \mathbf{B} applied in z for $0 T \leq \mathbf{B} \leq 3 T$ (upper panel) and $0 T \leq \mathbf{B} \leq 9 T$ (lower panel).”

- “Supplementary Fig. S13. Low-temperature Hall response in ZrTe_5 sample D. **a**, Longitudinal electrical resistivity ρ_{xx} (blue, left axis) and Hall resistivity ρ_{xy} (red, right axis) as a function of \mathbf{B} at $T = 50$ mK with \mathbf{B} applied in z for $0 T \leq \mathbf{B} \leq 3 T$ (upper panel) and $0 T \leq \mathbf{B} \leq 9 T$ (lower panel). The blue arrows mark the onset of a Landau level (LL). The blue and red numbers label the filling factor ν of the corresponding LL. **b**, Longitudinal electrical resistivity ρ_{xx} (blue, left axis) as a function of \mathbf{B} at $T = 50$ mK with \mathbf{B} applied in z for $0 T \leq \mathbf{B} \leq 3 T$ (upper panel) and $0 T \leq \mathbf{B} \leq 9 T$ (lower panel). **c**, Longitudinal electrical conductivity σ_{xx} (blue, left axis) and Hall conductivity σ_{xy} (red, right axis) as a function of \mathbf{B}^{-1} at $T = 50$ mK with \mathbf{B} applied in z . **d**, Variation of the longitudinal electrical resistivity $\Delta\rho_{xx}$ as a function of \mathbf{B} at 50 mK with \mathbf{B} applied along z (upper panel) and Landau-index fan diagram (lower panel) as a function of \mathbf{B}^{-1} at $T = 50$ mK with \mathbf{B} applied in z .”

- Supplementary Fig. S14. Low-temperature Hall response in ZrTe₅ sample E.** *a*, Longitudinal electrical resistivity ρ_{xx} (blue, left axis) and Hall resistivity ρ_{xy} (red, right axis) as a function of \mathbf{B} at $T = 50$ mK with \mathbf{B} applied in z for $0 T \leq \mathbf{B} \leq 3 T$ (upper panel) and $0 T \leq \mathbf{B} \leq 9 T$ (lower panel). The blue arrows mark the onset of a Landau level (LL). The blue and red numbers label the filling factor ν of the corresponding LL. *b*, Longitudinal electrical resistivity ρ_{xx} (blue, left axis) as a function of \mathbf{B} at $T = 50$ mK with \mathbf{B} applied in z for $0 T \leq \mathbf{B} \leq 3 T$ (upper panel) and $0 T \leq \mathbf{B} \leq 9 T$ (lower panel). *c*, Longitudinal electrical conductivity σ_{xx} (blue, left axis) and Hall conductivity σ_{xy} (red, right axis) as a function of \mathbf{B}^{-1} at $T = 50$ mK with \mathbf{B} applied in z . *d*, Variation of the longitudinal electrical resistivity $\Delta\rho_{xx}$ as a function of \mathbf{B} at 50 mK with \mathbf{B} applied along z (upper panel) and Landau-index fan diagram (lower panel) as a function of \mathbf{B}^{-1} at $T = 50$ mK with \mathbf{B} applied in z .

- Supplementary Fig. S15. Low-temperature Hall response in ZrTe₅ sample F.** *a*, Longitudinal electrical resistivity ρ_{xx} (blue, left axis) and Hall resistivity ρ_{xy} (red, right axis) as a function of \mathbf{B} at $T = 50$ mK with \mathbf{B} applied in z for $0 T \leq \mathbf{B} \leq 3 T$ (upper panel) and $0 T \leq \mathbf{B} \leq 9 T$ (lower panel). The blue arrows mark the onset of a Landau level (LL). The blue and red numbers label the filling factor ν of the corresponding LL. *b*, Longitudinal electrical resistivity ρ_{xx} (blue, left axis) as a function of \mathbf{B} at $T = 50$ mK with

\mathbf{B} applied in z for $0 T \leq \mathbf{B} \leq 3 T$ (upper panel) and $0 T \leq \mathbf{B} \leq 9 T$ (lower panel). **c.** Longitudinal electrical conductivity σ_{xx} (blue, left axis) and Hall conductivity σ_{xy} (red, right axis) as a function of \mathbf{B}^{-1} at $T = 50$ mK with \mathbf{B} applied in z . **d.** Variation of the longitudinal electrical resistivity $\Delta\rho_{xx}$ as a function of \mathbf{B} at 50 mK with \mathbf{B} applied along z (upper panel) and Landau-index fan diagram (lower panel) as a function of \mathbf{B}^{-1} at $T = 50$ mK with \mathbf{B} applied in z .

2. The authors believe their observation of "3D FQH" response in the extreme quantum limit regime. However, unlike the FQHE in a 2D electronic system, this observation is only appearing the "fractional 6/5 and 3/5" instead of the robust fractional "1/3" state in usual 2D system.

We understand that the main issue at this point is that we have observed the 6/5 and 3/5 plateau instead of the 1/3 state commonly found in 2D. Since in the previous section we have established that the 6/5 plateau is most likely originating from spin splitting in the following we will focus now only on the 3/5 state: Theoretical calculations investigating field induced phases in the quantum limit show a plethora of different possible ground states with nearly degenerate ground state energies (references: 9, 11, 12, 33, 34). This is perhaps best visualized when considering graphene bilayers where even denominator states $\nu = 1/2$ have been found even though they were not found in monolayer sheets. The interpretation presented in the discussed manuscript emerges from our related work, in which we have investigated the origin of the quantized plateaus in ZrTe_5 via studying its thermodynamic properties (ZrTe_5 (Galeski *et al.* arXiv:2005.12996)). In that work, we have found ZrTe_5 to be well modelled as a stack of weakly interacting Dirac 2DEGs. In *Phys. Rev. B* **42**, 1339 (1990) and *Phys. Rev. B* **37**, 4792 MacDonald *et al.* suggest that in such a setting a staging transition could take place, however it is pointed out that the exact ground state finely depends on the microscopic details in particular on the strength of the inter- and intra-layer coupling. The details of which remain an open question and need further experimental and theoretical investigation. To stress this point, we have now incorporated an open discussion of this issue in the main text and suggest experiments beyond the scope of this work to address this question:

- *"Although possible scenarios for an emergence a plateau in the Hall resistance in the quantum limit of electron plasma include formation of a CDW, Luttinger liquid, Wigner crystallization or the so called Hall crystal,^{35,36} the most likely origin stems from considering the notion that ZrTe_5 and HfTe_5 can be thought of as a stack of interacting 2DEGs. Based on a Hartree-Fock analysis it was proposed analysis⁹ that in a layered structure the gain in exchange energy can exceed the energy cost for distributing electrons unequally between layers. The electrons then undergo spontaneous staging transitions in which only every N -th layers is occupied, while all other layers are emptied (the number N depends on the average electron density and the state formed) – some of which are only stabilized due to the interplay of electron interaction and spin-orbit coupling.³⁷"*
- *"In conclusion, our measurements reveal a fractional plateau in the Hall conductivity of the bulk semimetal HfTe_5 at magnetic fields beyond the quantum limit. The plateau is accompanied by a Shubnikov-de Haas minimum in the longitudinal electrical resistivity. The height of the Hall plateaus is given by twice Fermi wave vector in the direction of the applied magnetic field and scales with $\nu = 3/5$ of the conductance quantum – signatures that are suggestive of a fractional Hall effect in a 3D crystal. The absence of the fractional plateau in isostructural single band ZrTe_5 samples with a similar electron mobility and Fermi wavevector, suggests the presence of a correlated state that may be stabilized by spin-orbit coupling. However, further experimental and theoretical efforts in determining the real interactions and texture of the Hall effect in HfTe_5 are necessary to settle the puzzle of the $\nu = 3/5$ -plateau. In particular experiments directly probing the*

Density of States and the real space charge distribution such as Scanning Tunnelling Spectroscopy and in-field X-ray diffraction could shed additional light on the nature of the observed feature.”

3. *To explain the 3D fractional composite fermions, it is essentially required to form a full energy gap with strong MANY-BODY interactions. However, I notice that the minima of $\rho_{xx}(B)$ is still finite (such as at the position of $\nu=6/5, 1, 3/5, \sim 3m\text{Ohms.cm}$, in Figure 3 (a)), and the authors interpreted it as 3D Hall effect with disorder in SI(S4). Those explanations are very strange to me, because the plateaus $\nu=1$ is attributed to the disorder (single particle picture, if I understand it correctly), whereas its front ($\nu=6/5$) and rear ($\nu=3/5$) regimes correspond to strong MANY-BODY interactions. This certainly requires further clarification, especially in the magnetic field direction.*

The described scenario after establishing the 6/5 as originating from spin splitting indeed assumes that below the quantum limit all physics can be at least qualitatively described within a single particle picture with disorder entering as a means of pinning the chemical potential (Galeski *et al.* arXiv:2005.12996). However, once the system enters the quantum limit this description is not sufficient in the case of HfTe₅. In a non-interacting system, one would expect a simple 1/B field dependence of the Hall resistance (as is the case for ZrTe₅). Thus, the emergence of robust features both in ρ_{xx} and ρ_{xy} requires an explanation beyond the single particle picture. Having said this, our data indeed displays a finite longitudinal resistivity in the vicinity of the 3/5 plateau. ρ_{xx} can be finite for a variety of reasons perhaps the simplest being competition between the different nearly degenerate ground states. Such an explanation, although evoking many-body effects, is not disconnected from effects of local disorder in a 3D bulk sample and is supported by the closeness of the high field insulating phase. In general, although in canonical QHE physics ρ_{xx} is expected to be zero, measurements on a variety of systems exhibiting the FQHE often display finite resistance in the vicinity of fractional plateaus (see for example: *Nature Physics* **7**, 693–696 (2011)).

Below are the changes made in the manuscript in order to address the raised issues:

- *We note, however, that our data displays a finite ρ_{xx} in the vicinity of the 3/5 plateau. This can occur for a variety of reasons perhaps the simplest being competition between the different nearly degenerate ground states. Such an explanation, although evoking many-body effects, is not disconnected from effects of local disorder in a 3D bulk sample and is supported by the closeness of the high field insulating phase. In general, although in canonical 2D QHE physics ρ_{xx} is expected to be zero, measurements on a variety of systems exhibiting the FQHE often display finite resistance in the vicinity of fractional plateaus (see for example ²⁹).*
4. *A comment in page 6 (line 139), the Hall resistivity tensor should be written as: $\rho_{xy} = \sigma_{xy} / (\sigma_{xx} \sigma_{yy} + \sigma_{xy}^2)$, and a short description of approximation needs to be added. Otherwise, it is quite misleading in an anisotropic system (since $\sigma_{xx}(B) \neq \sigma_{yy}(B)$).*

Thank you for pointing out this issue. In the original version of the manuscript, we have described the way we calculate the conductivity/resistivity tensors in the supplement. However, we agree that this discussion is of central importance and can cause confusion if not properly addressed. In order to make the presentation clearer, we have now included it in the main text:

- We note that the observed quantization of ρ_{xy} is not immediately obvious from the predicted quantization in σ_{xy} . The Hall resistivity tensor is given by $\rho_{xy} = \sigma_{xy} / (\sigma_{xx} \sigma_{yy} + \sigma_{xy}^2)$ with a magnetic field in z-direction, where σ_{xx} and σ_{yy} are the

longitudinal component of the conductivity tensor in x and y -direction, respectively. Vice versa, the Hall conductivity tensor element is given by $\sigma_{xy} = \rho_{xy}/(\rho_{xx}\rho_{yy} + \rho_{xy}^2)$ with a magnetic field in z -direction. However, in our samples at low temperatures $\sigma_{xx} < \sigma_{xy}$ (Supplementary Fig. S11 and Supplementary Fig. S12) and thus $\sigma_{xy}^{-1} \approx \rho_{xy}$, enabling the direct observation of the quantization. Due to the geometry of the HfTe₅ crystals (elongated needles) and their mechanical fragility, performing reliable measurements of ρ_{yy} is not possible. Instead, we estimate the error of the σ_{xy} using the ratio of Drude resistivities $\rho_{yy}/\rho_{xx} = (n_{xy}e^2\tau_x/m_x^*)/(n_{xy}e^2\tau_y/m_y^*)$ given by the quantum lifetimes and effective masses obtained from Shubnikov-de Haas oscillations on sample A (Supplementary Table S1). n_{xy} is the charge-carrier concentration in the x - y -plane. Based on this analysis we find $\rho_{yy}/\rho_{xx} \approx 0.4$. Owing to $\rho_{xx}(B) < \rho_{xy}(B)$ this results in an error of below 8 % in the estimated σ_{xy} for the investigated field range - well within the estimated error of $k_{F,z}$ of 10 %.

5. Page 5, line 103, there is a typo in pdf.

We would like to thank for the careful reading of our manuscript. Having revised the previous version, we have carried out additional proof reading in order to avoid such issues.

Again, we would like to thank you for your appreciation of our work and hope that in our response we were able to clarify all the issues to your satisfaction.

Sincerely yours,
Authors

Reviewer #2

Dear Reviewer,

thank you for your technical insight and rising the point of the flatness of the plateaus area. Indeed, we have not addressed these points sufficiently enough in the original manuscript. In the following, we respond to each point in detail, with changes made to the manuscript highlighted in blue.

1. *In line 133 and 134, the authors stated that “pronounced plateaus in Hall resistance.....features commonly related to the QHE (Fig 2a)”, but none of the plateaus in Fig. 2a is pronounced. Only the $\nu=1$ shoulder is plateau-like. The readers deserved to know how flat it is and the authors should provide a zoom-in. The Hall measurement in three-dimensional material is challenging, and a barely ideal plateau may also support the existence of $\nu=1$ integer quantum Hall state. Only based on the assumption of $\nu=1$ shoulder, one can guess the existence of $\nu=2$ state. Otherwise, a slope change in Hall resistance and an oscillation in longitudinal resistance around $1T$ are not as clear as that around $2.5T$.*

This point got indeed overlooked as the manuscript developed and we agree that showing a quantitative analysis of the flatness of the observed Hall plateaus is important. Hence, we now provide zoom-ins on the $\nu = 1$ and $\nu = 3/5$ Hall plateaus in the Supplementary Materials (Supplementary Fig. 17) and incorporated their discussion in the main text. Our analysis reveals that both the $\nu = 1$ and $\nu = 3/5$ plateaus are well developed in conductivity within 3% of the expected value in the range of 0.5 T around the plateau center. This makes the observed plateaus in HfTe_5 as flat as the established fractional quantum Hall plateaus observed in graphene (see for example *Nature* **462**, 196–199 (2009); *Science* **358**, 648-652 2017; *Nature Physics* **7**, 693–696 (2011); *Nature* **462**, 192–195(2009) and *Nature Nanotechnology* **12**, 118–122 (2017)) and multi-layer graphene systems (see for example *Science* **345**, 61-64 (2014) or *Science* **350**, 1231-1234 (2015)).

- *“Close investigation of the plateau height in conductivity reveals that both the $\nu = 1$ and $\nu = 3/5$ plateaus are well developed with conductivity within 3% of the expected value in the range of 0.5 T around the plateau center (Supplementary Fig. S16).”*

- **Supplementary Fig. S16. Flatness of the Hall plateaus of HfTe₅ sample A and B.** *a.* Hall conductivity σ_{xy} relative to $\sigma_{xy, \nu=1} = 1 \cdot \frac{k_{F,z} e^2}{\pi h}$ as a function of magnetic field B with respect to the center of the plateau B_{center} for Sample A at $T = 50$ mK with B applied in z . *b.* σ_{xy} relative to $\sigma_{xy, \nu=3/5} = 3/5 \cdot \frac{k_{F,z} e^2}{\pi h}$ as a function of B with respect to B_{center} for Sample A at $T = 50$ mK with B applied in z . *c.* σ_{xy} relative to $\sigma_{xy, \nu=1}$ as a function of B with respect to B_{center} for Sample B at $T = 50$ mK with B applied in z . *d.* σ_{xy} relative to $\sigma_{xy, \nu=3/5}$ as a function of B with respect to B_{center} for Sample B at $T = 50$ mK with B applied in z .

2. Then, if we focus on Fig. 3a, the slope changes labelled as 6/5 and 3/5 don't provide sufficient evidence to be fractional quantum Hall plateaus. There are other possible causes of the slope change in Hall resistance, and fractional quantum Hall effect is not the most straight forward explanation. Although I appreciate the efforts the authors have spent on this experiment, I still want to persuade the authors don't claim the fractional quantum response with the existing data. The observation of only the $\nu=1$ three-dimensional integer quantum Hall effect in HfTe₅ is probably a better conclusion.

As explained in response to the first comment, we have now quantitatively shown that the 3/5-plateau is as flat as fractional quantum Hall plateaus in graphene and graphene multilayer samples. However, we shared your concern that there are other possible causes of the slope change in Hall resistance. Therefore, we have tested whether the longitudinal resistance $\rho_{xx}(B)$ is connected to $\rho_{xy}(B)$ via $\rho_{xx}(B) = \gamma B \cdot d\rho_{xy}(B)/dB$, where γ is a dimensionless parameter of the order of 0.01-0.05 (see for example *Solid State Commun.* **56**, 153–154 (1985) and *Phys. Rev. Lett.* **73**, 3278 (1994)). This derivative relation is characteristic for quantum Hall physics in 2D systems for both FQHE and IQHE alike. In Fig. 4, we show that this derivative relation is well fulfilled with $\gamma = 0.04$ for our HfTe₅ samples. These results indicate that the observed plateaus observed in HfTe₅ are indeed be related to quantum Hall physics – including the $\nu = 3/5$.

In the main manuscript we write:

- “A related empirical observation is that in both FQHE and IQHE in 2D the longitudinal resistance $\rho_{xx}(B)$ is connected to $\rho_{xy}(B)$ via $\rho_{xx}(B) = \gamma B \cdot d\rho_{xy}(B)/dB$, where γ is a dimensionless parameter of the order of 0.01-0.05, which measures the local electron concentration fluctuations.^{32,33} Comparison of $\sigma_{xx}(B)$ (Fig. 4d) and $\gamma B \cdot d\rho_{xy}(B)/dB$ (Fig. 4e, upper panel) as a function of B^{-1} reveals that both quantities show maxima and minima at the same field positions as $\Delta\rho_{xx}(B)$. In particular, the derivative relation is well fulfilled with $\gamma = 0.04$, which is in the expected range reported for 2DEs. These results indicate that the observed $\nu = 3/5$ plateau observed in HfTe_5 is indeed be related to quantum Hall physics.”

Still, the origin of the 3/5- Hall plateau remains an open question. In order to verify whether the observed features can be explained by invoking the presence of a second pocket at the Fermi energy, we have performed magneto-transport measurements up to 70 T in the (Supplementary Fig. S17 and Ref. ²¹). The measurements did not reveal any additional quantum oscillations. Our data must, therefore, be analysed in terms of a single electron like Dirac pocket and, thus, single particle electron physics cannot account for the 3/5 plateau in the quantum limit.

As a cross check, we have now carried out additional experiments on isostructural ZrTe_5 samples of similar quality and Fermi vector for comparison. While this comparison reveals that the originally labelled 6/5 plateau most likely originates from spin splitting, it strengthens that the origin of the $\nu = 3/5$ lies beyond the single-particle interpretation (new Fig. 3 and corresponding changes in the manuscript). The additional data of the three ZrTe_5 samples measured is now included in the Supplementary Information and we have revised the manuscript accordingly with a focus on the observed 3/5-feature:

- “So far, our analysis focused on similarities between the IQHEs observed in ZrTe_5 and HfTe_5 . Upon cooling the samples to 50 mK, an obvious difference emerges, as shown in Fig. 4a investigated (Supplementary Fig. S11 - S15). At low fields below the quantum limit ($B < B_C$), both compounds exhibit signatures of new peaks and plateaus in $\rho_{xx}(B)$ and $\rho_{xy}(B)$. However, such features have been observed in the past and are related to spin splitting of the Landau levels.^{21,29,30} Interestingly, at high fields ($B > B_C$) – in the quantum limit, HfTe_5 exhibits an additional peak in $\rho_{xx}(B)$, accompanied by a plateau in $\rho_{xy}(B)$. This is in sharp contrast to ZrTe_5 , in which $\rho_{xx}(B)$ and $\rho_{xy}(B)$ smoothly increase. The additional maximum in HfTe_5 in the quantum limit is situated at the rational filling fraction $\nu = 3/5$ with regard to the first integer maximum at $N = 1$, which is identified using the Landau index fan diagram obtained with the integer fillings at 3 K. The $\nu = 3/5$ indexing is confirmed by corresponding maxima in $\rho_{xx}(B)$ and/or $\sigma_{xx}(B)$ of all three HfTe_5 samples investigated (compare Supplementary Fig. S11 and Supplementary Fig. S12), despite being less pronounced in some of them.”

• **Fig. 3. Low temperature longitudinal magnetoresistivity and Hall resistivity in isostructural HfTe₅ and ZrTe₅ at 50 mK.** **a,** Longitudinal electrical resistivity ρ_{xx} (blue, left axis) and Hall resistivity ρ_{xy} (red, right axis) of ZrTe₅ as a function of B/B_{OL} at $T = 50$ mK with the magnetic field \mathbf{B} applied in z for $0 T \leq B \leq 3 T$ (upper panel) and $0 T \leq B \leq 9 T$ (lower panel). The blue arrows mark $\nu = 3/5$ and $\nu = 1$ filling factors. B_{OL} denotes the magnetic field of the onset of the $\nu = 1$ Landau level. The blue and red numbers label the filling factor ν of the corresponding Landau level. **b,** Longitudinal electrical resistivity ρ_{xx} (blue, left axis) and Hall conductivity σ_{xy} (red, right axis) of ZrTe₅ as a function of B at $T = 50$ mK with \mathbf{B} applied in z for $0 T \leq B \leq 3 T$ (upper panel) and $0 T \leq B \leq 9 T$ (lower panel).

- **“Supplementary Fig. S13. Low-temperature Hall response in ZrTe₅ sample D. *a*, Longitudinal electrical resistivity ρ_{xx} (blue, left axis) and Hall resistivity ρ_{xy} (red, right axis) as a function of \mathbf{B} at $T = 50$ mK with \mathbf{B} applied in z for $0 T \leq \mathbf{B} \leq 3 T$ (upper panel) and $0 T \leq \mathbf{B} \leq 9 T$ (lower panel). The blue arrows mark the onset of a Landau level (LL). The blue and red numbers label the filling factor ν of the corresponding LL. *b*, Longitudinal electrical resistivity ρ_{xx} (blue, left axis) as a function of \mathbf{B} at $T = 50$ mK with \mathbf{B} applied in z for $0 T \leq \mathbf{B} \leq 3 T$ (upper panel) and $0 T \leq \mathbf{B} \leq 9 T$ (lower panel). *c*, Longitudinal electrical conductivity σ_{xx} (blue, left axis) and Hall conductivity σ_{xy} (red, right axis) as a function of \mathbf{B}^{-1} at $T = 50$ mK with \mathbf{B} applied in z . *d*, Variation of the longitudinal electrical resistivity $\Delta\rho_{xx}$ as a function of \mathbf{B} at 50 mK with \mathbf{B} applied along z (upper panel) and Landau-index fan diagram (lower panel) as a function of \mathbf{B}^{-1} at $T = 50$ mK with \mathbf{B} applied in z .”**

- Supplementary Fig. S14. Low-temperature Hall response in ZrTe_5 sample E.** *a*, Longitudinal electrical resistivity ρ_{xx} (blue, left axis) and Hall resistivity ρ_{xy} (red, right axis) as a function of \mathbf{B} at $T = 50$ mK with \mathbf{B} applied in z for $0 T \leq \mathbf{B} \leq 3 T$ (upper panel) and $0 T \leq \mathbf{B} \leq 9 T$ (lower panel). The blue arrows mark the onset of a Landau level (LL). The blue and red numbers label the filling factor ν of the corresponding LL. *b*, Longitudinal electrical resistivity ρ_{xx} (blue, left axis) as a function of \mathbf{B} at $T = 50$ mK with \mathbf{B} applied in z for $0 T \leq \mathbf{B} \leq 3 T$ (upper panel) and $0 T \leq \mathbf{B} \leq 9 T$ (lower panel). *c*, Longitudinal electrical conductivity σ_{xx} (blue, left axis) and Hall conductivity σ_{xy} (red, right axis) as a function of \mathbf{B}^{-1} at $T = 50$ mK with \mathbf{B} applied in z . *d*, Variation of the longitudinal electrical resistivity $\Delta\rho_{xx}$ as a function of \mathbf{B} at 50 mK with \mathbf{B} applied along z (upper panel) and Landau-index fan diagram (lower panel) as a function of \mathbf{B}^{-1} at $T = 50$ mK with \mathbf{B} applied in z .

- **Supplementary Fig. S15. Low-temperature Hall response in ZrTe₅ sample F.** *a*, Longitudinal electrical resistivity ρ_{xx} (blue, left axis) and Hall resistivity ρ_{xy} (red, right axis) as a function of \mathbf{B} at $T = 50$ mK with \mathbf{B} applied in z for $0 \leq \mathbf{B} \leq 3$ T (upper panel) and $0 \leq \mathbf{B} \leq 9$ T (lower panel). The blue arrows mark the onset of a Landau level (LL). The blue and red numbers label the filling factor ν of the corresponding LL. *b*, Longitudinal electrical resistivity ρ_{xx} (blue, left axis) as a function of \mathbf{B} at $T = 50$ mK with \mathbf{B} applied in z for $0 \leq \mathbf{B} \leq 3$ T (upper panel) and $0 \leq \mathbf{B} \leq 9$ T (lower panel). *c*, Longitudinal electrical conductivity σ_{xx} (blue, left axis) and Hall conductivity σ_{xy} (red, right axis) as a function of \mathbf{B}^{-1} at $T = 50$ mK with \mathbf{B} applied in z . *d*, Variation of the longitudinal electrical resistivity $\Delta\rho_{xx}$ as a function of \mathbf{B} at 50 mK with \mathbf{B} applied along z (upper panel) and Landau-index fan diagram (lower panel) as a function of \mathbf{B}^{-1} at $T = 50$ mK with \mathbf{B} applied in z .

Having excluded a single particle origin for the 3/5-feature, its ground state likely originates from correlation physics. Theoretical calculations investigating field induced phases in the quantum limit indeed show a plethora of different possible ground states with nearly degenerate ground state energies (references: 9, 11, 12, 33, 34). This is perhaps best visualized when considering graphene bilayers where even denominator states $\nu = 1/2$ have been found even though they were not found in monolayer sheets. The interpretation presented in the discussed manuscript emerges from our related work, in which we have investigated the origin of the quantized plateaus in ZrTe₅ via studying its thermodynamic properties (Galeski *et al.* arXiv:2005.12996). In that work, we have found ZrTe₅ to be well modelled as a stack of weakly interacting Dirac 2DEGs. In *Phys. Rev. B* **42**, 1339 (1990) and *Phys. Rev. B* **37**, 4792 MacDonald *et al.* suggests that in such a setting a staging transition could take place, however it is pointed out that the exact ground state finely depends on the microscopic details in particular on the strength of the inter- and intra-layer coupling. The details of which remain an open question and need further experimental and theoretical investigation. To stress this point, we have now incorporated a discussion of this issue in the main text and suggest experiments beyond the scope of this work:

- “*Interacting electrons in two dimensions can form exotic ground states, leading to the emergence of the fractional quantum Hall effect (FQHE). Although the FQHE has also been predicted to occur in three dimensions, it has not yet been experimentally observed. Here, we report the observation of a fractional plateau in the Hall conductivity of the bulk semimetal HfTe₅ at magnetic fields beyond the quantum limit. The plateau is accompanied by a Shubnikov-de Haas minimum in the longitudinal electrical resistivity. The height of the Hall plateau is given by twice the*

Fermi wave vector in the direction of the applied magnetic field and scales with the rational fraction 3/5 of the conductance quantum. Our findings are consistent with strong electron-electron interactions, stabilizing a fractionalized variant of the hall effect in three dimension...

- *“Although possible scenarios for an emergence a plateau in the Hall resistance in the quantum limit of electron gas include formation of a CDW, Luttinger liquid, Wigner crystallization or the so called Hall crystal,^{35,36} the most likely origin stems from considering the notion that ZrTe₅ and HfTe₅ can be thought of as a stack of interacting 2DEGs. Based on a Hartree-Fock analysis it was proposed analysis⁹ that in a layered structure the gain in exchange energy can exceed the energy cost for distributing electrons unequally between layers. The electrons then undergo spontaneous staging transitions in which only every N-th layers is occupied, while all other layers are emptied (the number N depends on the average electron density and the state formed) – some of which are only stabilized due to the interplay of electron interaction and spin-orbit coupling.³⁷”*
 - *“In conclusion, our measurements reveal a fractional plateau in the Hall conductivity of the bulk semimetal HfTe₅ at magnetic fields beyond the quantum limit. The plateau is accompanied by a Shubnikov-de Haas minimum in the longitudinal electrical resistivity. The height of the Hall plateaus is given by twice Fermi wave vector in the direction of the applied magnetic field and scales with $\nu = 3/5$ of the conductance quantum – signatures that are suggestive of a fractional Hall effect in a 3D crystal. The absence of the fractional plateau in isostructural single band ZrTe₅ samples with a similar electron mobility and Fermi wavevector, suggest the presence of a correlated state that may be stabilized by spin-orbit coupling. However, further experimental and theoretical efforts in determining the real interactions and texture of the Hall effect in HfTe₅ are necessary to settle the puzzle of the $\nu = 3/5$ -plateau. In particular experiments directly probing the Density of States and the real space charge distribution such as Scanning Tunnelling Spectroscopy and in-field X-ray diffraction could shed additional light on the nature of the observed feature.”*
3. *Last but not least, I have a technical concern. The longitudinal resistance corresponding to the expected plateaus should be zero or a dip. The authors intentionally labelled the filling factor at the peak position, which is much easier to see. Traditionally, the filling factor is labelled at the magnetic field of quantum Hall plateaus. A dip changing with temperature or a zero longitudinal resistance at the filling factor of plateau is much stronger evidence than a peak next to a plateau. At filling factors of $\nu=1$ and 2, there are dips, and the authors may check the temperature dependence to estimate the energy gap. At filling factors of $\nu=6/5$, there is no minimum in longitudinal resistance in Fig. 3a.*

Indeed, we label the Landau level index N and the corresponding filling factor ν at the peak position of the longitudinal resistivities, because traditionally the Landau index N is labelled in this way when extracting Landau index fan diagrams. However, we note that we did not sufficiently distinguish between N and ν in the previous version of the manuscript. To be more precise, we have now changed the notations in the figures and at the appropriate position of the manuscript, accordingly.

In addition, we have checked the temperature-dependence of the dips of the longitudinal resistivity and fitted the corresponding thermal activation energies. The results of which are shown in Supplementary Fig. S18 and discussed in the main text.

- Supplementary Fig. S18. Fit of the gap energies.** **a**, ρ_{xx} as a function of T^{-1} at the magnetic field B of the Shubnikov-de Haas minima of filling factor $\nu = 3/5$, **b**, $\nu = 1$, **c**, $\nu = 2$. The gap energies Δ_ν are fitted (red lines) in the thermally activated regime $\rho_{xx}(B) = \rho_{xx,A} \exp(\Delta_\nu/2k_B T) + \rho_{xx,0}$, where k_B is the Boltzmann constant, $\rho_{xx,A}$ a scaling factor and $\rho_{xx,0}$ accounts for the finite ρ_{xx} in the Shubnikov-de Haas minima due to disorder broadening of the Landau levels.
- Having established the relation of the observed features to QHE physics we have focused on gaining quantitative insights into the 3D Hall effect in HfTe₅. To do this, we have estimated the gap energies Δ_ν associated with the integer and fractional plateaus from the T -dependence of the $\rho_{xx}(B)$ minima (Fig. 4f) in the thermally activated regime $\rho_{xx}(B) \propto \exp(\Delta_\nu/2k_B T)$, where k_B is the Boltzmann constant (Supplementary Fig. S17). For integer ν , we find $\Delta_1 = (40 \pm 2)$ K at $\nu = 1$ and $\Delta_2 = (9 \pm 1)$ K at $\nu = 2$. The gap energies of the fractional plateau are two orders of magnitude lower: for the $\nu = 3/5$ state, we estimate $\Delta_{3/5} = (0.49 \pm 0.09)$ K. The deviations given for the gaps are the errors obtained from the thermally activated fits in Supplementary Fig. S18. In spite of considerable LL broadening, both the size of the gaps of the integer and fractional plateaus compare well with the gaps obtained in 2DESS.^{29,32} The different Δ_ν are also in agreement with the T -dependence of the corresponding Hall plateaus (Fig. 4g). While the integer plateaus are observable up to tens of Kelvin, the fractional ones vanish at around 0.5 K.

Again, we thank you for your comments and technical insight. We hope that the inclusion of the points raised in the review has improved the manuscript quality up to a standard that in your opinion would merit publication in *Nature Communications*.

Sincerely yours,
Authors

Reviewer #3

Dear Reviewer,

Thank you for your appreciation of our work. We are also deeply intrigued by the possibility of encountering 3D states affine to the canonical 2D fractional quantum Hall effect and hope our work could spark more experimental effort to shed light on their microscopic origin. In the following, we would like to address all concerns and inconsistencies that were pointed out in the review point by point, the changes made to the manuscript are highlighted in blue.

1. *The authors estimate the filling factor from the value of the Hall plateau, which they claim is confirmed by the resistance oscillations. According to Fig. 3a and 3e, the slope of the Hall is substantially reduced above the $n=1$ Landau level. This suggests that to arrive at the next plateau, for instance, $\nu=3/5$, one needs to go to a field higher than expected from the low field slope. Correspondingly, the oscillation period of the resistance will be expanded. So, the filling factors obtained by two methods should differ rather than agree.*

In the case of bare measured magnetoresistance this is indeed true that the slope of the Hall effect is reduced. However, we believe that this is the case due to the finite value of ρ_{xx} . A finite value of ρ_{xx} can originate from a variety of reasons perhaps the simplest being competition between the different nearly degenerate ground states (references: 9, 11, 12, 33, 34). Such a scenario seems very realistic for a bulk 3D sample that could manifest small local variations of electron density. In general, although in canonical QHE physics ρ_{xx} is expected to be zero, it is not uncommon for FQHE systems to display finite resistance in the vicinity of fractional plateaus (see for example: *Nature Physics* **7**, 693–696 (2011)).

Since theory of the QHE relates rather the current response to an applied electric field rather than the voltage drop in the presence of current we have inverted the resistivity tensor to compare the values of σ_{xy} directly. The comparison of the σ_{xy} value at the $3/5$ filling expected from longitudinal resistivity confirms that $\sigma_{xy, \nu=3/5}$ remains within 3% from the expected $\sigma_{xy, \nu=3/5} = 3/5 \cdot \frac{k_{F,z} e^2}{\pi h}$ within a 0.5 Tesla range. In order to clarify this, point we have included zoom-ins to the $\nu=1$ and $\nu=3/5$ plateaus.

- *“Close investigation of the plateau height in conductivity reveals that both the $\nu=1$ and $\nu=3/5$ plateaus are well developed with conductivity within 3% of the expected value in the range of 0.5 T around the plateau center (Supplementary Fig. S16).”*

- **Supplementary Fig. S16. Flatness of the Hall plateaus of HfTe₅ sample A and B.** *a.* Hall conductivity σ_{xy} relative to $\sigma_{xy, \nu=1} = 1 \cdot \frac{k_{F,z} e^2}{\pi h}$ as a function of magnetic field B with respect to the center of the plateau B_{center} for Sample A at $T = 50$ mK with B applied in z . *b.* σ_{xy} relative to $\sigma_{xy, \nu=3/5} = 3/5 \cdot \frac{k_{F,z} e^2}{\pi h}$ as a function of B with respect to B_{center} for Sample A at $T = 50$ mK with B applied in z . *c.* σ_{xy} relative to $\sigma_{xy, \nu=1}$ as a function of B with respect to B_{center} for Sample B at $T = 50$ mK with B applied in z . *d.* σ_{xy} relative to $\sigma_{xy, \nu=3/5}$ as a function of B with respect to B_{center} for Sample B at $T = 50$ mK with B applied in z .

Here, we would like to acknowledge the inherent error present in the analysis due to geometrical errors, uncertainty related to the extraction of Shubnikov-de Haas oscillations and additional error present due to the difficulty of inverting the resistivity tensor without detailed knowledge of ρ_{yy} . Nevertheless, considering the geometry error (the sample size is measured at various locations along the sample) and the error from ρ_{yy} (detailed description now in the main text), we estimate that the calculated value of σ_{xy} at $\nu=1$ and $\nu=3/5$ plateaus should have an error below 8% within the 10% error estimated for $3/5 \cdot \frac{k_{F,z} e^2}{\pi h}$ based on the analysis of Shubnikov-de Haas oscillations. What additionally convinces us of the accuracy of the procedure is the consistency of results between different samples. Including determination of the $\nu=1$ plateau heights of 3 ZrTe₅ samples, which data is now also included in the manuscript.

- *We note that the observed quantization of ρ_{xy} is not immediately obvious from the predicted quantization in σ_{xy} . The Hall resistivity tensor is given by $\rho_{xy} = \sigma_{xy} / (\sigma_{xx} \sigma_{yy} + \sigma_{xy}^2)$ with a magnetic field in z -direction, where σ_{xx} and σ_{yy} are the longitudinal component of the conductivity tensor in x and y -direction, respectively. Vice versa, the Hall conductivity tensor element is given by $\sigma_{xy} = \rho_{xy} / (\rho_{xx} \rho_{yy} + \rho_{xy}^2)$ with a magnetic field in z -direction. However, in our samples at low temperatures $\sigma_{xx} < \sigma_{yy}$ (Supplementary Fig. S11 and Supplementary Fig. S12) and thus $\sigma_{xy}^{-1} \approx \rho_{xy}$, enabling the direct observation of the quantization. Due to the geometry of the HfTe₅ crystals (elongated needles) and their mechanical fragility, performing reliable measurements of ρ_{yy} is not possible. Instead, we estimate the error of the σ_{xy} using the ratio of Drude resistivities $\rho_{yy} / \rho_{xx} = (n_{xy} e^2 \tau_x / m_x^*) / (n_{xy} e^2 \tau_y / m_y^*)$ given by the quantum lifetimes and*

effective masses obtained from Shubnikov-de Haas oscillations on sample A (Supplementary Table S1). n_{xy} is the charge-carrier concentration in the x - y -plane. Based on this analysis we find $\rho_{yy}/\rho_{xx} \approx 0.4$, which results in an error of below 8 % in the estimated σ_{xy} for the investigated field range owing to $\rho_{xx}(B) < \rho_{xy}(B)$. Both these errors lay within the estimated error of $k_{F,z}$ of 10 %.

2. For a Dirac dispersion, one anticipates a half-integer quantum Hall effect due to the non-trivial Berry phase, as seen in graphene. Taking into account the spin degeneracy, it is expected to see plateaus at filling factors of 1, 3, 5,... When Zeeman splitting is strong enough, even filling factors start to develop. For a parabolic dispersion, even filling factors should appear, that is, 2, 4, 6,... Odd filling factors will be produced by the Zeeman splitting at high fields. In this experiment, integer filling factors are equally spaced. One possibility is that the spin degeneracy is lifted for all observed Landau levels, even those occurring at low fields. This may not be easily justified. Note that previous study on ZrTe₅ have shown that Zeeman splitting of Landau levels appears only at high fields, *Nat. Commun.* 7, 12516 (2016). The second possibility is that the filling factors are in fact 2, 4, 6...instead of 1, 2, 3... Consequently, the fractional filling factors observed in this study could be due to the Zeeman splitting of $n=2$ and 4 Landau levels. Comparing the Hall slope and the SdH oscillation frequency may be able to tell whether the spin degeneracy is lifted.

We realize that this point would indeed not be straight forwardly explained within the original interpretation of the 3D QHE presented in the original work (*Nature* **569**, 537–541 (2019)). Here, however, we would like to recast the issue in the light of our most recent study of the origin of the 3D QHE in ZrTe₅ (<https://arxiv.org/abs/2005.12996>). In this work, we have shown that the appearance of the robust $\nu=1$ plateau originates from the combination of electron localization and the Dirac nature of electrons in ZrTe₅. This finding provides a very robust criterion for determining the $N=1$ Landau level.

Here we would like to point out that in the original manuscript we have considered the possibility of the 6/5 state as originating from spin splitting, but we had not addressed it in sufficient detail. In the current version of the manuscript, we have, therefore, included a 1:1 comparison between the low temperature Hall and longitudinal resistivity of ZrTe₅ and HfTe₅ (Figure 3 of the revised manuscript). The comparison has revealed that at low fields, below the quantum limit, both materials show almost the same behavior rendering the feature originally labeled at 6/5 most probably a signature of spin splitting.

However, significant differences appear in the quantum limit. For ZrTe₅, the Hall resistivity is as expected a smooth function of $1/B$, whereas for HfTe₅ a plateau like feature accompanied by a maximum in longitudinal resistivity is present at the 3/5 filling. Owing to the structural and electronic similarity between ZrTe₅ and HfTe₅, the robust high field feature in HfTe₅ has to be considered to emerge from a correlation effect stabilized by the increased spin-orbit coupling in HfTe₅. In fact, the appearance of an additional spin split level - the original 6/5 peak, further confirms that the strength of Zeeman effect is similar in both materials and thus precludes the origin of the 3/5 plateau from spin splitting. Below we list the changes made in the manuscript in order to address this issue:

- In ZrTe_5 and HfTe_5 , the IQHE was originally believed to arise from a charge density wave (CDW), due to the scaling of plateau height with the Fermi wavevector. This scenario is, however, in contrast with thermodynamic and thermoelectric measurements on ZrTe_5 that did not reveal any signatures of a field induced CDW transition. Instead, it was proposed that ZrTe_5 should be considered a stack of weakly interacting Dirac 2DEGs with the plateau height scaling originating from the interplay of extremely small carrier density and the peculiarities of Landau quantization of the Dirac dispersion.²¹ “So far, our

analysis focused on similarities between the IQHEs observed in ZrTe_5 and HfTe_5 . Upon cooling the samples to 50 mK, an obvious difference emerges, as shown in Fig. 4a investigated (Supplementary Fig. S11 - S14). At low fields below the quantum limit ($B < B_C$), both compounds exhibit signatures of new peaks and plateaus in $\rho_{xx}(B)$ and $\rho_{xy}(B)$. However, such features have been observed in the past and are related to spin splitting of the Landau levels.^{21,29}”

- “Fig. 3. Low temperature longitudinal magnetoresistivity and Hall resistivity in isostructural HfTe_5 and ZrTe_5 at 50 mK. **a**, Longitudinal electrical resistivity ρ_{xx} (blue, left axis) and Hall resistivity ρ_{xy} (red, right axis) of ZrTe_5 as a function of \mathbf{B}/B_{OL} at $T = 50$ mK with the magnetic field \mathbf{B} applied in z for $0 \text{ T} \leq \mathbf{B} \leq 3 \text{ T}$ (upper panel) and $0 \text{ T} \leq \mathbf{B} \leq 9 \text{ T}$ (lower panel). The blue arrows mark $\nu = 3/5$ and $\nu = 1$ filling factors. B_{OL} denotes the magnetic field of the onset of the $\nu = 1$ Landau level. The blue and red numbers label the filling factor ν of the corresponding Landau level. **b**, Longitudinal electrical resistivity ρ_{xx} (blue, left axis) and Hall conductivity σ_{xy} (red, right axis) of ZrTe_5 as a function of \mathbf{B} at $T = 50$ mK with \mathbf{B} applied in z for $0 \text{ T} \leq \mathbf{B} \leq 3 \text{ T}$ (upper panel) and $0 \text{ T} \leq \mathbf{B} \leq 9 \text{ T}$ (lower panel).”

- “**Supplementary Fig. S13. Low-temperature Hall response in ZrTe₅ sample D.** **a**, Longitudinal electrical resistivity ρ_{xx} (blue, left axis) and Hall resistivity ρ_{xy} (red, right axis) as a function of \mathbf{B} at $T = 50$ mK with \mathbf{B} applied in z for $0 T \leq \mathbf{B} \leq 3 T$ (upper panel) and $0 T \leq \mathbf{B} \leq 9 T$ (lower panel). The blue arrows mark the onset of a Landau level (LL). The blue and red numbers label the filling factor ν of the corresponding LL. **b**, Longitudinal electrical resistivity ρ_{xx} (blue, left axis) as a function of \mathbf{B} at $T = 50$ mK with \mathbf{B} applied in z for $0 T \leq \mathbf{B} \leq 3 T$ (upper panel) and $0 T \leq \mathbf{B} \leq 9 T$ (lower panel). **c**, Longitudinal electrical conductivity σ_{xx} (blue, left axis) and Hall conductivity σ_{xy} (red, right axis) as a function of \mathbf{B}^{-1} at $T = 50$ mK with \mathbf{B} applied in z . **d**, Variation of the longitudinal electrical resistivity $\Delta\rho_{xx}$ as a function of \mathbf{B} at 50 mK with \mathbf{B} applied along z (upper panel) and Landau-index fan diagram (lower panel) as a function of \mathbf{B}^{-1} at $T = 50$ mK with \mathbf{B} applied in z .”

- “**Supplementary Fig. S14. Low-temperature Hall response in ZrTe₅ sample E.** **a**, Longitudinal electrical resistivity ρ_{xx} (blue, left axis) and Hall resistivity ρ_{xy} (red, right axis) as a function of \mathbf{B} at $T = 50$ mK with \mathbf{B} applied in z for $0 T \leq \mathbf{B} \leq 3 T$ (upper panel) and $0 T \leq \mathbf{B} \leq 9 T$ (lower panel). The blue arrows mark the onset of a Landau level (LL). The blue and red numbers label the filling factor ν of the corresponding LL. **b**, Longitudinal electrical resistivity ρ_{xx} (blue, left axis) as a function of \mathbf{B} at $T = 50$ mK with \mathbf{B} applied in z for $0 T \leq \mathbf{B} \leq 3 T$ (upper panel) and $0 T \leq \mathbf{B} \leq 9 T$ (lower panel). **c**, Longitudinal electrical conductivity σ_{xx} (blue, left axis) and Hall conductivity σ_{xy} (red, right axis) as a function of \mathbf{B}^{-1} at $T = 50$ mK with \mathbf{B} applied in z . **d**, Variation of the longitudinal electrical resistivity $\Delta\rho_{xx}$ as a function of \mathbf{B} at 50 mK with \mathbf{B} applied along z (upper panel) and Landau-index fan diagram (lower panel) as a function of \mathbf{B}^{-1} at $T = 50$ mK with \mathbf{B} applied in z .”

- Supplementary Fig. S15. Low-temperature Hall response in $ZrTe_5$ sample F.** *a*, Longitudinal electrical resistivity ρ_{xx} (blue, left axis) and Hall resistivity ρ_{xy} (red, right axis) as a function of B at $T = 50$ mK with B applied in z for $0 T \leq B \leq 3 T$ (upper panel) and $0 T \leq B \leq 9 T$ (lower panel). The blue arrows mark the onset of a Landau level (LL). The blue and red numbers label the filling factor ν of the corresponding LL. *b*, Longitudinal electrical resistivity ρ_{xx} (blue, left axis) as a function of B at $T = 50$ mK with B applied in z for $0 T \leq B \leq 3 T$ (upper panel) and $0 T \leq B \leq 9 T$ (lower panel). *c*, Longitudinal electrical conductivity σ_{xx} (blue, left axis) and Hall conductivity σ_{xy} (red, right axis) as a function of B^{-1} at $T = 50$ mK with B applied in z . *d*, Variation of the longitudinal electrical resistivity $\Delta\rho_{xx}$ as a function of B at 50 mK with B applied along z (upper panel) and Landau-index fan diagram (lower panel) as a function of B^{-1} at $T = 50$ mK with B applied in z .

- A minor comment. In Fig. 3, all the arrows, red or blue, seem to represent the Landau level index n (the field at which a Landau level is aligned with the Fermi level) rather than the filling factor ν . It is confusing when these two are used interchangeably.

Thank you for this comment. We have now realized that the labeling could indeed be confusing. We have reworked all the figures accordingly in order to increase the clarity and readability.

Again, we thank your comments and insight into the possible effects of Zeeman splitting. We hope that a more careful treatment of the points raised in the review has improved the manuscript quality up and readability to a standard that in your opinion would merit publication in *Nature Communications*.

Sincerely yours,
Authors

REVIEWER COMMENTS

Reviewer #1 (Remarks to the Author):

In reply to referees' comments, the authors have satisfactorily addressed most of the comments. In this new version, they had explained their current observations of spin splitting state in 3D HfTe5 materials. My last concern is worrying about multi-pockets quantum oscillations scenario. In the independent ARPES measurement [Zhang Y, et al., Science Bulletin, 62(13),950,2017], the Fermi energy (around 10 meV) is likely cross multi-pockets (the main pocket is around Γ point, another pocket is closing to M point, in figure 2 (a) at 15K. This speculation can be further confirmed by first-principles calculation in ref. 25 (Fig. 8). I appreciated the authors providing the high magnetic field data (SI fig S17) up to 70Tesla. There is also simply explained as " $\nu=2$ and $\nu=1$ " by pocket in M point instead of " $\nu=6/5$ and $\nu=3/5$ " from Γ point pocket. Can authors distinguish between "fractional" and "multi-pockets" states?

In summary, if this major aspect is well explained. I will highly recommend proceeding with publication.

Reviewer #2 (Remarks to the Author):

In Fig. S16, the conductance at $3/5$ clearly changes monotonically with the magnetic field. It's risky to treat the conductance at $3/5$ as a plateau. The authors used literature from graphene as a support, but graphene is widely accepted as a two-dimensional electron gas and fractional quantum Hall effect is expected. The quantization of plateaus in graphene does not need any extra scalable parameter either. In the case of HfTe5, however, stronger experimental evidence is expected to convince the readers. In the literature provided by authors, Nature Physics 7, 693–696 (2011), resistance changes at $4/3$ and $7/3$ are clearly non-monotonic. In addition, the features appear as a group with the expected denominator 3. These made Cory Dean's claim of fractional quantum Hall effect in graphene reasonable. So far, the data presented here for $3/5$ is only a slope change, which can result from many different trivial reasons. A slope change in Hall resistance should not be considered as a plateau. The experimental evidence is not strong enough to guarantee the existence of fractional quantum Hall effect in this work.

The fitting in Fig. S18 is better plotted with y-axis in log-scale and with ρ_{xx0} subtracted, so that it is easier to compare the difference between the fitting and the data.

By the way, a minus sign seems to be missing from the equation of ρ_{xx} .

Reviewer #3 (Remarks to the Author):

The authors have carefully considered the comments raised by referees and revised the manuscript accordingly. Most of these concerns have been alleviated. The manuscript is now close to a form suitable for publication in Nature Comm. I have only one comment that I believe deserve clarification in the revision. It is related to the second comment in my first review report, but it has not been thoroughly considered. The filling factor ν is related to the Landau level index n as $\nu = g_s \cdot n$ for conventional electrons, or $\nu = g_s \cdot (n + 1/2)$ for Dirac electrons, where g_s is the spin degeneracy. Note that there is a factor of 2 difference. These two must be distinguished. In Fig. 4b and similar figures in the SM, the filling factor seems incorrectly labeled. Or it meant to be the Landau level index?

Reviewer #1 (Remarks to the Author):

In reply to referees' comments, the authors have satisfactorily addressed most of the comments. In this new version, they had explained their current observations of spin splitting state in 3D HfTe5 materials. My last concern is worrying about multi-pockets quantum oscillations scenario. In the independent ARPES measurement [Zhang Y, et al., Science Bulletin, 62(13),950,2017], the Fermi energy (around 10 meV) is likely cross multi-pockets (the main pocket is around Γ point, another pocket is closing to M point, in figure 2 (a) at 15K. This speculation can be further confirmed by first-principles calculation in ref. 25 (Fig. 8). I appreciated the authors providing the high magnetic field data (SI fig S17) up to 70Tesla. There is also simply explained as " $\nu=2$ and $\nu=1$ " by pocket in M point instead of " $\nu=6/5$ and $\nu=3/5$ " from Γ point pocket. Can authors distinguish between "fractional" and "multi-pockets" states?

In summary, if this major aspect is well explained. I will highly recommend proceeding with publication.

Dear Reviewer,

We agree that the issue of multiband quantum oscillations, such as from other Fermi pockets, is known and that its exclusion in our HfTe5 samples could be better quantitatively supported in our manuscript. Therefore, we have now estimated the Fermi energy for our samples from the quantum oscillations, which is (9 ± 2) meV. This Fermi level position is consistent with the ARPES study performed on our samples (Zhang Y et al., Science Bulletin, 62(13),950,2017) – the paper that you have mentioned. Our conclusion that there is only a single Fermi pocket at the Fermi level is also consistent with the band structure calculations (Ref. 26).

While apart from the hole/electron bands that have maxima/minima at the Γ point, there exist indeed additional side conduction bands close to the Fermi level. In Zhang Y et al., Science Bulletin, 62(13),950,2017 it is explicitly pointed out that "...The SCB (side conduction band) exhibits a similar shift downwards as CCB (center conduction band) with temperature decrease, but it always sits above the Fermi level at all temperatures we have measured...". At 15 K, this SCB is located ~ 5 meV above the Fermi level (lowest temperature measured in the ARPES study) as compared to the Fermi function broadening of $k_B \cdot 15 \text{ K} \approx 1$ meV. Below 15 K, the Fermi level stays constant with respect to the Fermi level as indicated by the constant Shubnikov-de Haas frequency and hence, the SCB cannot be observed by low-temperature transport experiments. We apologize for the lack of clarity on this point in the previous versions of the manuscript and have now added the discussion of which as a new section in the main text of the manuscript.

Importantly, we stress that we also see signatures of what was previously identified 6/5-state in ZrTe5, which we identify as spin splitting, but not for the 3/5-state in the ultra-quantum limit. Therefore, their observation cannot be simply explained simply as " $\nu=2$ and $\nu=1$ " originating from the pocket at the M point instead of " $\nu=6/5$ and $\nu=3/5$ " from Γ point pocket.

Again, we thank you for pointing out the possible issue of multiband effects. We hope that our quantitative analysis and further explanations clarified that there is no experimental indication for additional bands in our HfTe5 samples at the Fermi level.

Sincerely yours,

Authors

Reviewer #2 (Remarks to the Author):

In Fig. S16, the conductance at $3/5$ clearly changes monotonically with the magnetic field. It's risky to treat the conductance at $3/5$ as a plateau. The authors used literature from graphene as a support, but graphene is widely accepted as a two-dimensional electron gas and fractional quantum Hall effect is expected. The quantization of plateaus in graphene does not need any extra scalable parameter either. In the case of HfTe5, however, stronger experimental evidence is expected to convince the readers. In the literature provided by authors, Nature Physics 7, 693–696 (2011), resistance changes at $4/3$ and $7/3$ are clearly non-monotonic. In addition, the features appear as a group with the expected denominator 3. These made Cory Dean's claim of fractional quantum Hall effect in graphene reasonable. So far, the data presented here for $3/5$ is only a slope change, which can result from many different trivial reasons. A slope change in Hall resistance should not be considered as a plateau. The experimental evidence is not strong enough to guarantee the existence of fractional quantum Hall effect in this work.

The fitting in Fig. S18 is better plotted with y-axis in log-scale and with ρ_{xx0} subtracted, so that it is easier to compare the difference between the fitting and the data.

By the way, a minus sign seems to be missing from the equation of ρ_{xx} .

Dear Reviewer,

Thank you for your critical assessment of our manuscript. At this point, we tend to agree that a strong statement about fractional states could be considered not evident from the presented data. Although we still find the explanation of the plateau like feature to most likely originate from a staging transition as envisaged by Mc Donald et al., we agree that our data does not directly exclude other possible origins.

Nevertheless, we stand by our point that the observed plateau-like feature that appears only at very low temperatures would be consistent with a correlation-driven change in the electronic ground state. The possibility of such field induced states of matter has been long postulated, however, experimental evidence for their existence is thus far scarce. In addition, in many systems that show such signatures interpretation of the data is difficult due to the intrinsic complexities of the band structure (multiple band, fermi pockets of different carrier type etc.).

The simple band structure and the possibility of directly comparing ZrTe5 and HfTe5 described in our manuscript opens avenues for further research into the physics of electrons in the quantum limit. In this sense we believe that results presented in our manuscript are an important experimental step in understanding physics of interacting electrons in the quantum limit.

In order to address the issue of interpretation of our data, we have significantly reworked our manuscript in order to tone down our original claim of FQHE in 3D. Instead, now we have focused on the physics of electrons in the lowest Landau level. In addition, as suggested we have replotted Fig S18 for better readability.

Again, we would like to thank you for your assessment of our work. We hope that you will find the modified version of the manuscript to be up to the standard that would merit publication in Nature Communications.

Sincerely yours,

Authors

Reviewer #3 (Remarks to the Author):

The authors have carefully considered the comments raised by referees and revised the manuscript accordingly. Most of these concerns have been alleviated. The manuscript is now close to a form suitable for publication in Nature Comm. I have only one comment that I believe deserve clarification in the revision. It is related to the second comment in my first review report, but it has not been thoroughly considered. The filling factor ν is related to the Landau level index n as $\nu = g_s \cdot n$ for conventional electrons, or $\nu = g_s \cdot (n + 1/2)$ for Dirac electrons, where g_s is the spin degeneracy. Note that there is a factor of 2 difference. These two must be distinguished. In Fig. 4b and similar figures in the SM, the filling factor seems incorrectly labeled. Or it meant to be the Landau level index?

Dear Reviewer,

Thank you for raising this point again. We have noted that compared to most Landau level analyses in 3D bulk samples, in which the Landau level index notation is $N \geq 1$, in Dirac 2DEGs like Graphene the notation is rather $N \geq 0$. Hence, our notation for the massive Dirac systems can be confusing, if not explicitly explained. In contrast to massless Dirac systems, the gap-mass in HfTe5 and ZrTe5 splits the last Landau level into a spin-degenerated electron and a hole band and shifts them away from zero energy.

In two dimensions, the filling factor coincides with the number of Landau levels below the chemical potential. In three dimensions, where Landau levels form bands dispersing parallel to the field, the filling factor is actually not precisely defined anymore. We have, therefore, now removed all notions of a filling from the manuscript and the SI and refer now to the magnitude of the Hall-feature in the quantum limit of HfTe5 always with respect to the last plateau of the last integer Landau level.

Indeed, in Fig. 4b and similar figures in the Supporting Information it is meant to be the Landau level index N .

Again, we thank you for pointing out this issue. We have reworked the figures and the text accordingly to be consistent and increase the clarity and readability of our manuscript.

Sincerely yours,

Authors

REVIEWERS' COMMENTS

Reviewer #1 (Remarks to the Author):

In page 9, line 206, I suggest removing the word "must"; The corresponding revision has met the review comments, I then recommend it to be published in Nature communications.

Reviewer #2 (Remarks to the Author):

I support the publication of this manuscript.

Reviewer #3 (Remarks to the Author):

The authors have considered my comments and revised the manuscript accordingly. I now recommend its publication.

REVIEWER COMMENTS

Reviewer #1 (Remarks to the Author):

In page 9, line 206, I suggest removing the word “must”; The corresponding revision has met the review comments, I then recommend it to be published in Nature communications.

Dear Reviewer,

Thank you for the appreciation of our work and your recommendation to publish it in Nature Communications.

In the current revision we have followed your advice and replaced the word ‘must’. In the current manuscript version we have replaced the original sentence: ‘*Our data must, therefore, be analysed in terms of a single electron-type Dirac pocket.*’ with ‘*Our data can, therefore, be analysed in terms of a single electron-type Dirac pocket.*’

Again, we would like to thank you for your comments that in our view have significantly improved the quality of our manuscript.

Sincerely yours,

Authors

Reviewer #2 (Remarks to the Author):

I support the publication of this manuscript.

Dear Reviewer,

we would like to thank you for your critical assessment of our manuscript. We believe that your comments and constructive criticism has helped us substantially improve the quality of our manuscript and helped us sharpen our line of argumentation.

Again we would like to thank you for the appreciation of our work and your recommendation to publish it in Nature Communications.

Sincerely yours,

Authors

Reviewer #3 (Remarks to the Author):

The authors have considered my comments and revised the manuscript accordingly. I now recommend its publication.

Dear Reviewer,

Thank you for your comments and critical assessment of the nomenclature used throughout the manuscript. We believe that your comments have helped us greatly to improve the presentation clarity and readability of our manuscript.

In addition we would like to thank you for the appreciation of our work and your recommendation to publish it in Nature Communications.

Sincerely yours,

Authors